# Non-asymptotic Confidence Intervals of Off-policy Evaluation: Primal and Dual Bounds

**Yihao Feng** [*], **Ziyang Tang** [*]
University of Texas at Austin
`{yihao, ztang}@cs.utexas.edu`

**Na Zhang**
Tsinghua University
`zhangna@pbcsf.tsinghua.edu.cn`

**Qiang Liu**
University of Texas at Austin
`lqiang@cs.utexas.edu`

## Abstract

Off-policy evaluation (OPE) is the task of estimating the expected reward of a given policy based on offline data previously collected under different policies. Therefore, OPE is a key step in applying reinforcement learning to real-world domains such as medical treatment, where interactive data collection is expensive or even unsafe. As the observed data tends to be noisy and limited, it is essential to provide rigorous uncertainty quantification, not just a point estimation, when applying OPE to make high stakes decisions. This work considers the problem of constructing non-asymptotic confidence intervals in infinite-horizon off-policy evaluation, which remains a challenging open question. We develop a practical algorithm through a primal-dual optimization-based approach, which leverages the kernel Bellman loss (KBL) of Feng et al. (2019) and a new martingale concentration inequality of KBL applicable to time-dependent data with unknown mixing conditions. Our algorithm makes minimum assumptions on the data and the function class of the Q-function, and works for the behavior-agnostic settings where the data is collected under a mix of arbitrary unknown behavior policies. We present empirical results that clearly demonstrate the advantages of our approach over existing methods.

## 1 Introduction

Off-policy evaluation (OPE) seeks to estimate the expected reward of a target policy in reinforcement learnings (RL) from observational data collected under different policies (e.g., Murphy et al., 2001; Fonteneau et al., 2013; Jiang & Li, 2016; Liu et al., 2018a). OPE plays a central role in applying reinforcement learning (RL) with only observational data and has found important applications in areas such as medicine, self-driving, where interactive "on-policy" data is expensive or even infeasible to collect. A critical challenge in OPE is the uncertainty estimation, as having reliable confidence bounds is essential for making high-stakes decisions. In this work, we aim to tackle this problem by providing non-asymptotic confidence intervals of the expected value of the target policy. Our method allows us to rigorously quantify the uncertainty of the prediction and hence avoid the dangerous case of being overconfident in making costly and/or irreversible decisions.

However, off-policy evaluation per se has remained a key technical challenge in the literature (e.g., Precup, 2000; Thomas & Brunskill, 2016; Jiang & Li, 2016; Liu et al., 2018a), let alone gaining rigorous confidence estimation of it. This is especially true when 1) the underlying RL problem is long or infinite horizon, and 2) the data is collected under arbitrary and unknown algorithms (a.k.a. behavior-agnostic). As a consequence, the collected data can exhibit arbitrary dependency structure, which makes constructing rigorous non-asymptotic confidence bounds particularly challenging. Traditionally, the only approach to provide non-asymptotic confidence bounds in OPE is to combine importance sampling (IS) with concentration inequalities (e.g., Thomas et al., 2015a;b), which, however, tends to degenerate for long/infinite horizon problems (Liu et al., 2018a). Furthermore,

---

[*]Equal contribution.

neither can this approach be applied to the behavior-agnostic settings, nor can it effectively handle the complicated time dependency structure inside individual trajectories. Instead, it requires to use a large number of independently collected trajectories drawn under known policies.

In this work, we provide a practical approach for **B**ehavior-agnostic, **O**ff-policy, **I**nfinite-horizon, **N**on-asymptotic, **C**onfidence intervals based on arbitrarily **D**ependent data (**BONDIC**). Our method is motivated by a recently proposed optimization-based (or variational) approach to estimating OPE confidence bounds (Feng et al., 2020), which leverages a tail bound of kernel Bellman statistics (Feng et al., 2019). Our approach achieves a new bound that is both an order-of-magnitude tighter and computationally efficient than that of Feng et al. (2020). Our improvements are based on two pillars 1) developing a new primal-dual perspective on the non-asymptotic OPE confidence bounds, which is connected to a body of recent works on infinite-horizon value estimation (Liu et al., 2018a; Nachum et al., 2019a; Tang et al., 2020a; Mousavi et al., 2020); and 2) offering a new tight concentration inequality on the kernel Bellman statistics that applies to behavior-agnostic off-policy data with arbitrary dependency between transition pairs. Empirically, we demonstrate that our method can provide reliable and tight bounds on a variety of well-established benchmarks.

**Related Work**   Besides the aforementioned approach based on the combination of IS and concentration inequalities (e.g., Thomas et al., 2015a), bootstrapping methods have also been widely used in off-policy estimation (e.g., White & White, 2010; Hanna et al., 2017; Kostrikov & Nachum, 2020). But the latter is limited to asymptotic bounds. Alternatively, Bayesian methods (e.g. Engel et al., 2005; Ghavamzadeh et al., 2016a) offers a different way to estimate the uncertainty in RL, but fails to guarantee frequentist coverage. In addition, Distributed RL (Bellemare et al., 2017) seeks to quantify the intrinsic uncertainties inside the Markov decision process, which is orthogonal to the estimation of uncertainty that we consider.

Our work is built upon the recent advances in behavior-agnostic infinite-horizon OPE, including Liu et al. (2018a); Feng et al. (2019); Tang et al. (2020a); Mousavi et al. (2020), as well as the DICE-family (e.g., Nachum et al., 2019a; Zhang et al., 2020a; Yang et al., 2020b). In particular, our method can be viewed as extending the minimax framework of the infinite-horizon OPE in the *infinite data region* by Tang et al. (2020a); Uehara et al. (2020); Jiang & Huang (2020) to the *non-asymptotic* finite sample region.

**Outline**   For the rest of the paper, we start with the problem statement in Section 2 , and an overview on the two dual approaches to infinite-horizon OPE that are tightly connected to our method in Section 3. We then present our main approach in Section 4 and perform empirical studies in Section 5. The proof and an abundance of additional discussions can be found in Appendix.

## 2   BACKGROUND, DATA ASSUMPTION, PROBLEM SETTING

Consider an agent acting in an unknown environment. At each time step $t$, the agent observes the current state $s_t$ in a state space $\mathcal{S}$, takes an action $a_t \sim \pi(\cdot \mid s_t)$ in an action space $\mathcal{A}$ according to a *given* policy $\pi$; then, the agent receives a reward $r_t$ and the state transits to $s'_t = s_{t+1}$, following an *unknown* transition/reward distribution $(r_t, s_{t+1}) \sim \mathsf{P}(\cdot \mid s_t, a_t)$. Assume the initial state $s_0$ is drawn from an *known* initial distribution $\mathsf{D}_0$. Let $\gamma \in (0, 1)$ be a discount factor. In this setting, the expected reward of $\pi$ is defined as $J_\pi := \mathbb{E}_\pi \left[ \sum_{t=0}^{T} \gamma^t r_t \mid s_0 \sim \mathsf{D}_0 \right]$, which is the expected total discounted rewards when we execute $\pi$ starting from $\mathsf{D}_0$ for $T$ steps. In this work, we consider the *infinite-horizon* case with $T \to +\infty$.

Our goal is to provide an interval estimation of $J_\pi$ for a general and challenging setting with significantly released constraints on the data. In particular, we assume the data is *behavior-agnostic* and *off-policy*, which means that the data can be collected from multiple experiments, each of which can execute a mix of arbitrary, unknown policies, or even follow a non-fixed policy. More concretely, suppose that the model $\mathsf{P}$ is unknown, and we have a set of transition pairs $\hat{\mathsf{D}}_n = (s_i, a_i, r_i, s'_i)_{i=1}^n$ collected from previous experiments in a sequential order, such that for each data point $i$, the $(r_i, s'_i)$ is drawn from the model $\mathsf{P}(\cdot \mid s_i, a_i)$, while $(s_i, a_i)$ is generated with an arbitrary black box given the previous data points. We formalize both the data assumption and goal as below.

**Assumption 2.1** (**Data Assumption**). *Assume the data $\hat{\mathsf{D}}_n = (s_i, a_i, r_i, s_i')_{i=1}^n$ is drawn from an arbitrary joint distribution, such that for each $i = 1, \ldots, n$, conditional on $\hat{\mathsf{D}}_{<i} := (s_j, a_j, r_j, s_j')_{j<i} \cup (s_i, a_i)$, the subsequent local reward and next state $(r_i, s_i')$ are drawn from $\mathsf{P}(\cdot \mid s_i, a_i)$.*

**Goal** *Given a confidence level $\delta \in (0, 1)$, we want to construct an interval $[\hat{J}^-, \hat{J}^+] \subset \mathbb{R}$ based on the data $\hat{\mathsf{D}}_n$, such that $\Pr(J_\pi \in [\hat{J}^-, \hat{J}^+]) \geq 1 - \delta$, where $\Pr(\cdot)$ is w.r.t. the randomness of the data.*

The partial ordering on the data points is introduced to accommodate the case that $s_{i+1}$ equals $s_j'$ for some $j \leq i$. The data assumption only requires that $(r_i, s_i')$ is generated from $\mathsf{P}(\cdot \mid s_i, a_i)$, and imposes no constraints on how $(s_i, a_i)$ is generated. This provides great flexibility in terms of the data collection process. In particular, we do not require $(s_i, a_i)_{i=1}^n$ to be independent as always assumed in recent works (Liu et al., 2018a; Mousavi et al., 2020).

A crucial fact is that our data assumption actually implies a martingale structure on the empirical Bellman residual operator of the Q-function, As we will show in Section 4.1, this enables us to derive a key concentration inequality underpinning our non-asymptotic confidence bounds.

Here, we summarize a few notations that will simplify the presentation in the rest of work. First of all, we append each $(s_i, a_i, r_i, s_i')$ with an action $a_i' \sim \pi(\cdot \mid s_i')$ following $s_i'$. This can be done for free as long as $\pi$ is given (See the Remark in Section 3). Also, we write $x_i = (s_i, a_i)$, $x_i' = (s_i', a_i')$, and $y_i = (x_i', r_i) = (s_i', a_i', r_i)$. Correspondingly, define $\mathcal{X} = \mathcal{S} \times \mathcal{A}$ to be the state-action space and $\mathcal{Y} = \mathcal{X} \times \mathbb{R}$. Denote $\mathsf{P}_\pi(y \mid x) = \mathsf{P}(s', r \mid x)\pi(a' \mid s')$. In this way, the observed data can be written as pairs of $\{x_i, y_i\}_{i=1}^n$, and Assumption 2.1 is equivalent to saying that $y_i \sim \mathsf{P}_\pi(\cdot \mid x_i)$ given $\hat{\mathsf{D}}_{<i}$, which is similar to a supervised learning setting. We equalize the data $\hat{\mathsf{D}}_n$ with its empirical measure $\hat{\mathsf{D}}_n = \sum_{i=1}^n \delta_{x_i, y_i}/n$, where $\delta$ is the Delta measure.

## 3 Two dual approaches to Infinite-horizon off-policy estimation

The deficiency of the traditional IS methods on long-/infinite-horizon RL problems (a.k.a. *the curse of horizon* (Liu et al., 2018a)) has motivated a line of work on developing efficient infinite-horizon value estimation (e.g., Liu et al., 2018a; Feng et al., 2019; Nachum et al., 2019a; Zhang et al., 2020a; Mousavi et al., 2020; Tang et al., 2020a). The main idea is to transform the value estimation problem into estimating either the *Q-function* or the *visitation distribution (or its related density ratio)* of the policy $\pi$. This section introduces and reinterprets these two tightly connected methods, which serves to lay out a foundation for our main confidence bounds from a primal and dual perspective.

Given a policy $\pi$, its Q-function is defined as $q_\pi(x) = \mathbb{E}_\pi\left[\sum_{t=0}^\infty \gamma^t r_t \mid x_0 = x\right]$, where the expectation is taken when we execute $\pi$ initialized from a fixed state-action pair $(s_0, a_0) = x_0 = x$. Let $\mathsf{D}_{\pi,t}$ be the distribution of $(x_t, y_t) = (s_t, a_t, s_t', a_t', r_t)$ when executing policy $\pi$ starting from $s_0 \sim \mathsf{D}_0$ for $t$ steps. The visitation distribution of $\pi$ is defined as $\mathsf{D}_\pi = \sum_{t=0}^\infty \gamma^t \mathsf{D}_{\pi,t}$. Note that $\mathsf{D}_\pi$ integrates to $1/(1-\gamma)$, while we treat it as a probability measure in the notation.

The expected reward $J_\pi$ can be expressed using either $q_\pi$ or $\mathsf{D}_\pi$ as follows:

$$J_\pi := \mathbb{E}_\pi\left[\sum_{t=0}^\infty \gamma^t r_t\right] = \mathbb{E}_{r \sim \mathsf{D}_\pi}[r] = \mathbb{E}_{x \sim \mathsf{D}_{\pi,0}}[q_\pi(x)], \tag{1}$$

where $r \sim \mathsf{D}_\pi$ (resp. $x \sim \mathsf{D}_{\pi,0}$) denotes sampling from the $r$-(resp. $x$-) marginal distribution of $\mathsf{D}_\pi$ (resp. $\mathsf{D}_{\pi,0}$). Eq. (1) plays a key role in the infinite-horizon value estimation by transforming the estimation of $J_\pi$ into estimating either $q_\pi$ or $\mathsf{D}_\pi$.

**Value Estimation via Q Function** Because $\mathsf{D}_{\pi,0}(x) = \mathsf{D}_0(s)\pi(a|s)$ is known, we can estimate $J_\pi$ by $\mathbb{E}_{x \sim \mathsf{D}_{\pi,0}}[\hat{q}(x)]$ with any estimation $\hat{q}$ of the true Q-function $q_\pi$; the expectation under $x \sim \mathsf{D}_{\pi,0}$ can be estimated to any accuracy with Monte Carlo. To estimate $q_\pi$, we consider the empirical and expected Bellman residual operator:

$$\hat{\mathbf{R}}q(x, y) = q(x) - \gamma q(x') - r, \qquad \mathbf{R}_\pi q(x) = \mathbb{E}_{y \sim \mathsf{P}_\pi(\cdot|x)}\left[\hat{\mathbf{R}}q(x, y)\right]. \tag{2}$$

It is well-known that $q_\pi$ is the unique solution of the *Bellman equation* $\mathbf{R}_\pi q = 0$. Since $y_i \sim \mathsf{P}_\pi(\cdot|x_i)$ for each data point in $\hat{\mathsf{D}}_n$, if $q = q_\pi$, then $\hat{\mathbf{R}}q(x_i, y_i)$, $i = 1, \ldots, n$ are all *zero-mean* random variables.

Let $\omega$ be any function from $\mathcal{X}$ to $\mathbb{R}$, then $\sum_i \hat{\mathbf{R}}q(x_i, y_i)\omega(x_i)$ also has zero mean. This motivates the following *functional* Bellman loss (Feng et al., 2019; 2020; Xie & Jiang, 2020),

$$L_{\mathcal{W}}(q;\ \hat{\mathsf{D}}_n) := \sup_{\omega \in \mathcal{W}} \left\{ \frac{1}{n} \sum_{i=1}^n \hat{\mathbf{R}}q(x_i, y_i)\omega(x_i) \right\}, \tag{3}$$

where $\mathcal{W}$ is a set of functions $\omega\colon \mathcal{X} \to \mathbb{R}$. To ensure that the sup is finite, $\mathcal{W}$ is typically set to be an unit ball of some normed function space $\mathcal{W}_o$, such that $\mathcal{W} = \{\omega \in \mathcal{W}_o\colon \|\omega\|_{\mathcal{W}_o} \leq 1\}$. Feng et al. (2019) considers the simple case when $\mathcal{W}$ is taken to be the unit ball $\mathcal{K}$ of the reproducing kernel Hilbert space (RKHS) with a positive definite kernel $k\colon \mathcal{X} \times \mathcal{X} \to \mathbb{R}$, in which case the loss has a simple closed form solution:

$$L_{\mathcal{K}}(q;\ \hat{\mathsf{D}}_n) = \sqrt{\frac{1}{n^2} \sum_{ij=1}^n \hat{\mathbf{R}}q(x_i, y_i)k(x_i, x_j)\hat{\mathbf{R}}q(x_j, y_j)}. \tag{4}$$

Note that the RHS of Eq. (4) is the *square root* of the kernel Bellman V-statistics in Feng et al. (2019). Feng et al. (2019) showed that, when the support of data distribution $\hat{\mathsf{D}}_n$ covers the whole space (which may require an infinite data size) and $k$ is an integrally strictly positive definite kernel, $L_{\mathcal{K}}(q;\ \hat{\mathsf{D}}_n) = 0$ iff $q = q_\pi$. Therefore, one can estimate $q_\pi$ by minimizing $L_{\mathcal{K}}(q, \hat{\mathsf{D}}_n)$.

**Remark** The empirical Bellman residual operator $\hat{\mathbf{R}}$ can be extended to $\hat{\mathbf{R}}q(x, y) = q(x) - r - \gamma\frac{1}{m}\sum_{\ell=1}^m q(s', a'_\ell)$, where $\{a'_\ell\}_{i=1}^m$ are i.i.d. drawn from $\pi(\cdot|s')$. As $m$ increases, this gives a lower variance estimation of $\mathbf{R}_\pi q$. If $m = +\infty$, we have $\hat{\mathbf{R}}q(x, y) = q(x) - r - \gamma\mathbb{E}_{a'\sim\pi(\cdot\ |\ s')}[q(s', a')]$, which coincides with the operator used in the expected SARSA (Sutton & Barto, 1998). In fact, without any modification, all results in this work can be applied to $\hat{\mathbf{R}}q$ for any $m$.

**Value Estimation via Visitation Distribution** Another way to estimate $J_\pi$ in Eq. (1) is to approximate $\mathsf{D}_\pi$ with a weighted empirical measure of the data (Liu et al., 2018a; Nachum et al., 2019a; Mousavi et al., 2020; Zhang et al., 2020a). The key idea is to assign an importance weight $\omega(x_i)$ to each data point $x_i$ in $\hat{\mathsf{D}}_n$. We can choose the function $\omega\colon \mathcal{X} \to \mathbb{R}$ properly such that $\mathsf{D}_\pi$ and hence $J_\pi$ can be approximated by the $\omega$-weighted empirical measure of $\hat{\mathsf{D}}_n$ as follows:

$$J_\pi \approx \hat{J}_\omega := \mathbb{E}_{\hat{\mathsf{D}}_n^\omega}[r] = \frac{1}{n}\sum_{i=1}^n \omega(x_i)r_i, \qquad \mathsf{D}_\pi \approx \hat{\mathsf{D}}_n^\omega := \frac{1}{n}\sum_{i=1}^n \omega(x_i)\delta_{x_i, y_i}. \tag{5}$$

Intuitively, $\omega$ can be viewed as the density ratio between $\mathsf{D}_\pi$ and $\hat{\mathsf{D}}_n$, although the empirical measure $\hat{\mathsf{D}}_n$ may not have well-defined density. Liu et al. (2018a); Mousavi et al. (2020) proposed to estimate $\omega$ by minimizing a discrepancy measure between $\hat{\mathsf{D}}_n^\omega$ and $\mathsf{D}_\pi$. To see this, note that $\mathsf{D} = \mathsf{D}_\pi$ if and only if $\Delta(\mathsf{D}, q) = 0$ for any function $q$, where

$$\Delta(\mathsf{D}, q) = \mathbb{E}_{\mathsf{D}}[\gamma q(x') - q(x)] - \mathbb{E}_{\mathsf{D}_\pi}[\gamma q(x') - q(x)]$$
$$= \mathbb{E}_{\mathsf{D}}[\gamma q(x') - q(x)] + \mathbb{E}_{\mathsf{D}_{\pi,0}}[q(x)], \tag{6}$$

using the fact that $\mathbb{E}_{\mathsf{D}_\pi}[\gamma q(x') - q(x)] = -\mathbb{E}_{\mathsf{D}_{\pi,0}}[q(x)]$ (Theorem 1, Liu et al., 2018a). Also note that the RHS of Eq. (6) can be practically calculated given any $\mathsf{D}$ and $q$ without knowing $\mathsf{D}_\pi$. Let $\mathcal{Q}$ be a set of functions $q\colon \mathcal{X} \to \mathbb{R}$. One can define the following loss for $\omega$:

$$I_{\mathcal{Q}}(\omega;\ \hat{\mathsf{D}}_n) = \sup_{q\in\mathcal{Q}} \left\{ \Delta(\hat{\mathsf{D}}_n^\omega, q) \right\}. \tag{7}$$

Similar to $L_{\mathcal{W}}(q;\ \hat{\mathsf{D}}_n)$, when $\mathcal{Q}$ is a ball in RKHS, $I_{\mathcal{Q}}(\omega;\ \hat{\mathsf{D}}_n)$ also has a bilinear closed form analogous to Eq. (4); see Mousavi et al. (2020) and Appendix F. As we show in Section 4, $I_{\mathcal{Q}}(\omega;\ \hat{\mathsf{D}}_n)$ and $L_{\mathcal{W}}(q;\ \hat{\mathsf{D}}_n)$ are connected to the primal and dual views of our confidence bounds, respectively.

## 4 MAIN APPROACH

Let $\mathcal{Q}$ be a large enough function set including the true Q-function $q_\pi$, that is, $q_\pi \in \mathcal{Q}$. Following Feng et al. (2020), a confidence interval $[\hat{J}_{\mathcal{Q},\mathcal{W}}^-, \hat{J}_{\mathcal{Q},\mathcal{W}}^+]$ of $J_\pi$ can be constructed as follows:

$$\hat{J}_{\mathcal{Q},\mathcal{W}}^+ = \sup_{q\in\mathcal{Q}} \left\{ \mathbb{E}_{\mathsf{D}_{\pi,0}}[q]\ \ s.t.\ \ L_{\mathcal{W}}(q;\ \hat{\mathsf{D}}_n) \leq \varepsilon_n \right\}, \tag{8}$$

and $\hat{J}_{\mathcal{Q},\mathcal{W}}^-$ is defined in a similar way by replacing $\sup$ on $q \in \mathcal{Q}$ with $\inf$.

The idea here is to seek the extreme $q$ function with the largest (resp. smallest) expected values in set $\mathcal{F} := \mathcal{Q} \cap \{q \colon L_{\mathcal{K}}(q; \hat{\mathsf{D}}_n) \le \varepsilon_n\}$. Therefore, Eq. (8) would be a $1 - \delta$ confidence interval if $q_\pi$ is included in $\mathcal{F}$ with at least probability $1 - \delta$, which is ensured when $q_\pi \in \mathcal{Q}$ and

$$\Pr(L_{\mathcal{W}}(q_\pi; \hat{\mathsf{D}}_n) \le \varepsilon_n) \ge 1 - \delta\,. \tag{9}$$

Feng et al. (2020) showed that in the RKHS case when $\mathcal{W} = \mathcal{K}$, Eq. (9) can be achieved with

$$\varepsilon_n = \sqrt{2 c_{q_\pi, k} \left( \frac{n-1}{n} \sqrt{\frac{\log(1/\delta)}{n}} + \frac{1}{n} \right)}\,, \tag{10}$$

when $n$ is an even number, where $c_{q_\pi, k} = \sup_{x,y} \hat{\mathbf{R}} q_\pi(x,y)^2 k(x,x)$. This was proved using Hoeffding's inequality for U-statistics (Hoeffding, 1963). To solve Eq. (8) efficiently, Feng et al. (2020) took $\mathcal{Q}$ to be a ball in RKHS with random feature approximation. Unfortunately, this method as described by Eq. (8)-(10) has two major disadvantages:

**1) Bound Needs to Be Tightened (Section 4.1)**  The bound of $\varepsilon_n = O(n^{-1/4})$ in Eq. (10) is sub-optimal in rate. In Section 4.1, we improve it by an $\varepsilon_n = O(n^{-1/2})$ bound under the mild Assumption 2.1, which gets rid of the independence requirement between the transition pairs. Our tightened bound is achieved by firstly noting a Martingale structure on the empirical Bellman operator under Assumption 2.1, and then applying an inequality in Pinelis (1992).

**2) Dependence on Global Optimization (Section 4.2)**  The bound in Eq. (8) is guaranteed to be a $1 - \delta$ confidence bound only when the maximization in Eq. (8) is solved to global optimality. With a large $n$, this leads to a high computational cost, even when choosing $\mathcal{Q}$ as the RKHS. Feng et al. (2020) solved Eq. (8) approximately using a random feature technique, but this method suffers from a gap between the theory and practice. In Section 4.2, we address this problem by presenting a dual form of Eq. (8), which sidesteps solving the challenging global optimization in Eq. (8). Moreover, the dual form enables us to better analyze the tightness of the confidence interval and issues regarding the choices of $\mathcal{Q}$ and $\mathcal{W}$.

## 4.1 A Tighter Concentration Inequality

In this section, we explain our method to improve the bound in Eq. (10) by giving a tighter concentration inequality for the kernel Bellman loss in Eq. (4). We introduce the following *semi-expected* kernel Bellman loss:

$$L_{\mathcal{K}}^*(q; \hat{\mathsf{D}}_n) = \sqrt{\frac{1}{n^2} \sum_{ij=1}^{n} \mathbf{R}_\pi q(x_i) k(x_i, x_j) \mathbf{R}_\pi q(x_j)}\,, \tag{11}$$

in which we replace the empirical Bellman residual operator $\hat{\mathbf{R}} q$ in Eq. (3) with its expected counterpart $\mathbf{R}_\pi q$, but still take the empirical average over $\{x_i\}_{i=1}^n$ in $\hat{\mathsf{D}}_n$. For a more general function set $\mathcal{W}$, we can similarly define $L_{\mathcal{W}}^*(q; \hat{\mathsf{D}}_n)$ by replacing $\hat{\mathbf{R}} q$ with $\mathbf{R}_\pi q$ in Eq. (3). Obviously, we have $L_{\mathcal{W}}^*(q; \hat{\mathsf{D}}_n) = 0$ when $q = q_\pi$.

Theorem 4.1 below shows that $L_{\mathcal{K}}(q; \hat{\mathsf{D}}_n)$ concentrates around $L_{\mathcal{K}}^*(q; \hat{\mathsf{D}}_n)$ with an $O(n^{-1/2})$ error under Assumption 2.1. At a first glance, it may seem surprising that the concentration bound is able to hold even without any independence assumption between $\{x_i\}$. An easy way to make sense of this is by recognizing that the randomness in $y_i$ conditional on $x_i$ is aggregated through averaging, even when $\{x_i\}$ are deterministic. As Assumption 2.1 does not impose any (weak) independence between $\{x_i\}$, we cannot establish that $L_{\mathcal{K}}(q; \hat{\mathsf{D}}_n)$ concentrates around its mean $\mathbb{E}_{\hat{\mathsf{D}}_n}[L_{\mathcal{K}}(q; \hat{\mathsf{D}}_n)]$ (which is a *full* expected kernel bellman loss), without introducing further assumptions.

**Theorem 4.1.** *Assume $\mathcal{K}$ is the unit ball of RKHS with a positive definite kernel $k(\cdot, \cdot)$. Let $c_{q,k} := \sup_{x \in \mathcal{X}, y \in \mathcal{Y}} (\hat{\mathbf{R}} q(x,y) - \mathbf{R}_\pi q(x))^2 k(x,x) < \infty$. Under Assumption 2.1, for any $\delta \in (0,1)$, with at*

*least probability* $1 - \delta$, *we have*

$$\left| L_{\mathcal{K}}(q;\ \hat{\mathsf{D}}_n) - L_{\mathcal{K}}^*(q;\ \hat{\mathsf{D}}_n) \right| \leq \sqrt{\frac{2c_{q,k}\log(2/\delta)}{n}} \, . \tag{12}$$

*In particular, when* $q = q_\pi$, *we have* $c_{q_\pi,k} = \sup_{x,y}(\hat{\mathbf{R}}q_\pi(x,y))^2 k(x,x)$, *and*

$$L_{\mathcal{K}}(q_\pi;\ \hat{\mathsf{D}}_n) \leq \sqrt{\frac{2c_{q_\pi,k}\log(2/\delta)}{n}} \, . \tag{13}$$

Intuitively, to see why we can expect an $O(n^{-1/2})$ bound, note that $L_{\mathcal{K}}(q, \hat{\mathsf{D}}_n)$ consists of the square root of the product of two $\hat{\mathbf{R}}q$ terms, each of which contributes an $O(n^{-1/2})$ error w.r.t. $\mathbf{R}_\pi q$.

Technically, the proof is based on a key observation: Assumption 2.1 ensures that $Z_i := \hat{\mathbf{R}}q(x_i, y_i) - \mathbf{R}_\pi q(x_i)$, $i = 1, \ldots, n$ forms a *martingale difference* sequence w.r.t. $\{\hat{\mathsf{D}}_{<i} \colon \forall i = 1, \ldots, n\}$, in the sense that $\mathbb{E}[Z_i \mid \hat{\mathsf{D}}_{<i}] = 0, \forall i$. See Appendix B for details. The proof also leverages a special property of RKHS and applies a Hoeffding-like inequality on the Hilbert spaces as in Pinelis (1992) (see Appendix B). For other more general function sets $\mathcal{W}$, we establish in Appendix E a similar bound by using Rademacher complexity, although it yields a less tight bound than Eq. (12) when $\mathcal{W} = \mathcal{K}$.

## 4.2 DUAL CONFIDENCE BOUNDS

We derive a dual form of Eq. (8) that sidesteps the need for solving the challenging global optimization in Eq. (8). To do so, let us plug the definition of $L_{\mathcal{W}}(q;\ \hat{\mathsf{D}}_n)$ into Eq. (3) and introduce a Lagrange multiplier:

$$\hat{J}_{\mathcal{Q},\mathcal{W}}^+ = \sup_{q \in \mathcal{Q}} \inf_{h \in \mathcal{W}} \inf_{\lambda \geq 0} \mathbb{E}_{\mathsf{D}_{\pi,0}}[q] - \lambda \left( \frac{1}{n} \sum_{i=1}^n h(x_i)\hat{\mathbf{R}}q(x_i, y_i) - \varepsilon_n \right) \tag{14}$$

$$= \sup_{q \in \mathcal{Q}} \inf_{\omega \in \mathcal{W}_o} \left\{ \mathbb{E}_{\mathsf{D}_{\pi,0}}[q] - \frac{1}{n} \sum_{i=1}^n \omega(x_i)\hat{\mathbf{R}}q(x_i) + \varepsilon_n \|\omega\|_{\mathcal{W}_o} \right\} , \tag{15}$$

where we use $\omega(x) = \lambda h(x)$. Exchanging the order of min/max and some further derivation yields the following main result.

**Theorem 4.2.** *I) Let* $\mathcal{W}$ *be the unit ball of a normed function space* $\mathcal{W}_o$. *We have*

$$\begin{aligned}
\hat{J}_{\mathcal{Q},\mathcal{W}}^+ &\leq \hat{F}_{\mathcal{Q}}^+(\omega) := \mathbb{E}_{\hat{\mathsf{D}}_n^\omega}[r] + I_{\mathcal{Q}}(\omega;\ \hat{\mathsf{D}}_n) + \varepsilon_n \|\omega\|_{\mathcal{W}_o}, \quad \forall \omega \in \mathcal{W}_o, \\
\hat{J}_{\mathcal{Q},\mathcal{W}}^- &\geq \hat{F}_{\mathcal{Q}}^-(\omega) := \mathbb{E}_{\hat{\mathsf{D}}_n^\omega}[r] - I_{-\mathcal{Q}}(\omega;\ \hat{\mathsf{D}}_n) - \varepsilon_n \|\omega\|_{\mathcal{W}_o}, \quad \forall \omega \in \mathcal{W}_o,
\end{aligned} \tag{16}$$

*where* $-\mathcal{Q} = \{-q \colon q \in \mathcal{Q}\}$ *and hence* $I_{-\mathcal{Q}}(\omega;\ \hat{\mathsf{D}}_n) = I_{\mathcal{Q}}(\omega;\ \hat{\mathsf{D}}_n)$ *if* $\mathcal{Q} = -\mathcal{Q}$. *Further, we have* $\hat{J}_{\mathcal{Q},\mathcal{W}}^+ = \inf_{\omega \in \mathcal{W}_o} \hat{F}_{\mathcal{Q}}^+(\omega)$ *and* $\hat{J}_{\mathcal{Q},\mathcal{W}}^- = \sup_{\omega \in \mathcal{W}_o} \hat{F}_{\mathcal{Q}}^-(\omega)$ *if* $\mathcal{Q}$ *is convex and there exists a* $q \in \mathcal{Q}$ *that satisfies the strict feasibility condition that* $L_{\mathcal{W}}(q;\ \hat{\mathsf{D}}_n) < \varepsilon_n$.

*II) For* $\hat{\mathsf{D}}_n$ *and* $\delta \in (0, 1)$, *assume* $\mathcal{W}_o$ *and* $\varepsilon_n \in \mathbb{R}$ *satisfy Eq. (9) (e.g., via Theorem 4.1). Then for any function set* $\mathcal{Q}$ *with* $q_\pi \in \mathcal{Q}$, *and any function* $\omega_+, \omega_- \in \mathcal{W}_o$ *(the choice of* $\mathcal{Q}$, $\omega_+$, $\omega_-$ *can depend on* $\hat{\mathsf{D}}_n$ *arbitrarily), we have*

$$\Pr\left( J_\pi \in \left[ \hat{F}_{\mathcal{Q}}^-(\omega_-),\ \hat{F}_{\mathcal{Q}}^+(\omega_+) \right] \right) \geq 1 - \delta \, . \tag{17}$$

Theorem 4.2 transforms the original bound in Eq. (8), framed in terms of $q$ and $L_{\mathcal{W}}(q;\ \hat{\mathsf{D}}_n)$, into a form that involves the density-ratio $\omega$ and the related loss $I_{\mathcal{Q}}(\omega;\ \hat{\mathsf{D}}_n)$. The bounds in Eq. (16) can be interpreted as assigning an error bar around the $\omega$-based estimator $\hat{J}_\omega = \mathbb{E}_{\hat{\mathsf{D}}_n^\omega}[r]$ in Eq. (5), with the error bar of $I_{\pm\mathcal{Q}}(\omega;\ \hat{\mathsf{D}}_n) + \varepsilon_n \|\omega\|_{\mathcal{W}_o}$. Specifically, the first term $I_{\pm\mathcal{Q}}(\omega;\ \hat{\mathsf{D}}_n)$ measures the discrepancy between $\hat{\mathsf{D}}_n^\omega$ and $\mathsf{D}_\pi$ as discussed in Eq. (7), whereas the second term captures the randomness in the empirical Bellman residual operator $\hat{\mathbf{R}}q_\pi$.

Compared with Eq. (8), the global maximization on $q \in \mathcal{Q}$ is now transformed inside the $I_{\mathcal{Q}}(\omega; \hat{\mathsf{D}}_n)$ term, which yields a simple closed form solution in the RKHS case (see Appendix F). In practice, we can optimize $\omega_+$ and $\omega_-$ to obtain the tightest possible bound (and hence recover the primal bound) by minimizing/maximizing $\hat{F}_{\mathcal{Q}}^+(\omega)$ and $\hat{F}_{\mathcal{Q}}^-(\omega)$, but it is not necessary to solve the optimization to global optimality. When $\mathcal{W}_o$ is an RKHS, by the standard finite representer theorem (Scholkopf & Smola, 2018), the optimization on $\omega$ reduces to a finite dimensional optimization, which can be sped up with any favourable approximation techniques. We elaborate on this in Appendix D.

**Length of the Confidence Interval**   The form in Eq. (16) also makes it much easier to analyze the tightness of the confidence interval. Suppose $\omega = \omega_+ = \omega_-$ and $\mathcal{Q} = -\mathcal{Q}$, the length of the optimal confidence interval is

$$\text{length}([\hat{J}_{\mathcal{Q},\mathcal{W}}^-, \ \hat{J}_{\mathcal{Q},\mathcal{W}}^+]) = \inf_{\omega \in \mathcal{W}_o} \left\{ 2I_{\mathcal{Q}}(\omega; \ \hat{\mathsf{D}}_n) + 2\varepsilon_n \|\omega\|_{\mathcal{W}_o} \right\}.$$

Given $\varepsilon_n$ is $O(n^{-1/2})$, we can make the overall length of the optimal confidence interval also $O(n^{-1/2})$, as long as $\mathcal{W}_o$ is rich enough to include a *good* density ratio estimator $\omega^*$ that satisfies $I_{\mathcal{Q}}(\omega^*; \ \hat{\mathsf{D}}_n) = O(n^{-1/2})$ and has a bounded norm $\|\omega^*\|_{\mathcal{W}_o}$.

We can expect to achieve $I_{\mathcal{Q}}(\omega^*; \ \hat{\mathsf{D}}_n) = O(n^{-1/2})$, when (1) $\mathcal{Q}$ has an $O(n^{-1/2})$ sequential Rademacher complexity (Rakhlin et al., 2015) (e.g., a finite ball in RKHS); and (2) $\hat{\mathsf{D}}_n$ is collected following a Markov chain with strong mixing condition and weakly converges to some limit distribution $\mathsf{D}_\infty$ whose support is $\mathcal{X}$, and therefore we can define $\omega^*$ as the density ratio between $\mathsf{D}_\pi$ and $\mathsf{D}_\infty$. Refer to Appendix C for more discussions. Indeed, our experiments show that the lengths of practically constructed confidence intervals do tend to decay with an $O(n^{-1/2})$ rate.

**Choice of $\mathcal{W}$ and $\mathcal{Q}$**   To ensure the concentration inequality in Theorem 4.1 is valid, the choice of $\mathcal{W}_o$ cannot depend on the data $\hat{\mathsf{D}}_n$. Therefore, we should use a separate holdout data to construct a data-dependent $\mathcal{W}_o$. In contrast, *the choice of $\mathcal{Q}$ can depend on the data $\hat{\mathsf{D}}_n$ arbitrarily*, since it is a part of the optimization bound Eq. (8) but not in the tail bound Eq. (9). In this light, one can construct the best possible $\mathcal{Q}$ by exploiting the data information in the most favourable way. For example, we can construct an estimator of $\hat{q} \approx q_\pi$ based on any state-of-the-art method (e.g., Q-learning or model-based methods), and set $\mathcal{Q}$ to be a ball centering around $\hat{q}$ such that $q_\pi - \hat{q} \in \mathcal{Q}$. This enables *post-hoc analysis* based on prior information on $q_\pi$, as suggested in Feng et al. (2020).

**Mis-specification of $\mathcal{Q}$ and Oracle Upper/Lower Estimates**   Our result relies on the assumption that $q_\pi \in \mathcal{Q}$. However, as with other statistical estimation problems, there exists no provably way to empirically verify the correctness of model assumptions such as $q_\pi \in \mathcal{Q}$. Because empirical data only reveals the information of the unknown function (in our case $q_\pi$) on a finite number data points, but no conclusion can be made on the unseeing data points without imposing certain smoothness assumption. Typically, what we can do is the opposite: reject $q_\pi \in \mathcal{Q}$ when the Bellman loss $L_{\mathcal{W}}(q; \ \hat{\mathsf{D}}_n)$ of all $q$ in $\mathcal{Q}$ is larger than the threshold $\varepsilon_n$.

We highlight that, even without verifying $q_\pi \in \mathcal{Q}$, our method can still be viewed as a confidence interval of a best possible (oracle) upper and lower estimation given the data $\hat{\mathsf{D}}_n$ *plus* the assumption that $q_\pi \in \mathcal{Q}$, defined as

$$\hat{J}_{\mathcal{Q},*}^+ = \sup_{q \in \mathcal{Q}} \left\{ \mathbb{E}_{\mathsf{D}_{\pi,0}}[q] \quad s.t. \quad \hat{\mathbf{R}}q(x_i, y_i) = \hat{\mathbf{R}}q_\pi(x_i, y_i), \quad \forall i = 1, \ldots, n \right\}. \tag{18}$$

In fact, it is impossible to derive empirical upper bounds lower than $\hat{J}_{\mathcal{Q},*}^+$, as there is no way to distinguish $q$ and $q_\pi$ if $\hat{\mathbf{R}}q(x_i, y_i) = \hat{\mathbf{R}}q_\pi(x_i, y_i)$ for all $i$. But our interval $[\hat{J}_{\mathcal{Q},\mathcal{K}}^-, \hat{J}_{\mathcal{Q},\mathcal{K}}^+]$ provides a $1 - \delta$ confidence outer bound of $[\hat{J}_{\mathcal{Q},*}^-, \hat{J}_{\mathcal{Q},*}^+]$ once Eq. (9) holds, regardless if $q_\pi \in \mathcal{Q}$ holds or not. Hence, it is of independent interest to further explore the dual form of Eq. (18), which is another starting point for deriving our bound. We have more discussion in Appendix G.

Lastly, we argue that it is important to include the $\mathcal{Q}$ in the bound. Proposition G.1 in Appendix shows that removing the $q \in \mathcal{Q}$ constraint in Eq. (18) would lead to an infinite upper bound,

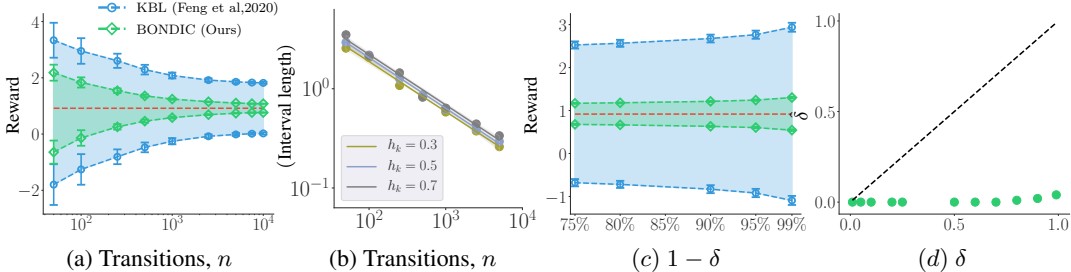

<table>
(a) Transitions, $n$      (b) Transitions, $n$      (c) $1 - \delta$      (d) $\delta$
</table>

Figure 1: Results on Inverted-Pendulum. (a) The confidence interval (significance level $\delta = 0.1$) of our method (green) and that of Feng et al. (2020) (blue) when varying the data size $n$. (b) The length of the confidence intervals ($\delta = 0.1$) of our method scaling with the data size $n$. (c) The confidence intervals when we vary the significance level $\delta$ (data size $n = 5000$). (d) The significance level $\delta$ vs. the empirical failure rate $\hat{\delta}$ of capturing the true expected reward by our confidence intervals (data size $n = 5000$). We average over 50 random trials for each experiment.

unless the $\{s_i, s_i'\}_{i=1}^n$ from $\hat{D}_n$ almost surely covers the whole state space $\mathcal{S}$ in the sense that $\Pr_{s \sim D_0}(s \in \{s_i, s_i'\}_{i=1}^n) = 1$.

## 5 EXPERIMENTS

We compare our method with a variety of existing algorithms for obtaining asymptotic and non-asymptotic bounds on a number of benchmarks. We find our method can provide confidence interval that correctly covers the true expected reward with probability larger than the specified success probability $1 - \delta$ (and is hence safe) across the multiple examples we tested. In comparison, the non-asymptotic bounds based on IS provide much wider confidence intervals. On the other hand, the asymptotic methods, such as bootstrap, despite giving tighter intervals, often fail to capture the true values with the given probability in practice.

**Environments and Dataset Construction** We test our method on three environments: Inverted-Pendulum and CartPole from OpenAI Gym (Brockman et al., 2016), and a Type-1 Diabetes medical treatment simulator.[1] We follow a similar procedure as Feng et al. (2020) to construct the behavior and target policies. more details on environments and data collection procedure are included in Appendix H.1.

**Algorithm Settings** We test the dual bound described in our paper. Throughout the experiment, we always set $\mathcal{W} = \mathcal{K}$, the unit ball of the RKHS with positive definite kernel $k$, and set $\mathcal{Q} = r_{\mathcal{Q}}\tilde{\mathcal{K}}$, the ball of radius $r_{\mathcal{Q}}$ in the RKHS with another kernel $\tilde{k}$. We take both kernels to be Gaussian RBF kernel and choose $r_{\mathcal{Q}}$ and the bandwidths of $k$ and $\tilde{k}$ using the procedure in Appendix H.2. We use a fast approximation method to optimize $\omega$ in $F_{\mathcal{Q}}^+(\omega)$ and $F_{\mathcal{Q}}^-(\omega)$ as shown in Appendix D. Once $\omega$ is found, we evaluate the bound in Eq. (16) exactly to ensure that the theoretical guarantee holds.

**Baseline Algorithms** We compare our method with four existing baselines, including the IS-based non-asymptotic bound using empirical Bernstein inequality by Thomas et al. (2015b), the IS-based bootstrap bound of Thomas (2015), the bootstrap bound based on fitted Q evaluation (FQE) by Kostrikov & Nachum (2020), and the bound in Feng et al. (2020) which is equivalent to the primal bound in (8) but with looser concentration inequality (they use a $\varepsilon_n = O(n^{-1/4})$ threshold).

**Results** Figure 1 shows our method obtains much tighter bounds than Feng et al. (2020), which is because we use a much tighter concentration inequality, even the dual bound that we use can be slightly looser than the primal bound used in Feng et al. (2020). Our method is also more computationally efficient than that of Feng et al. (2020) because the dual bound can be tightened

---

[1] https://github.com/jxx123/simglucose.

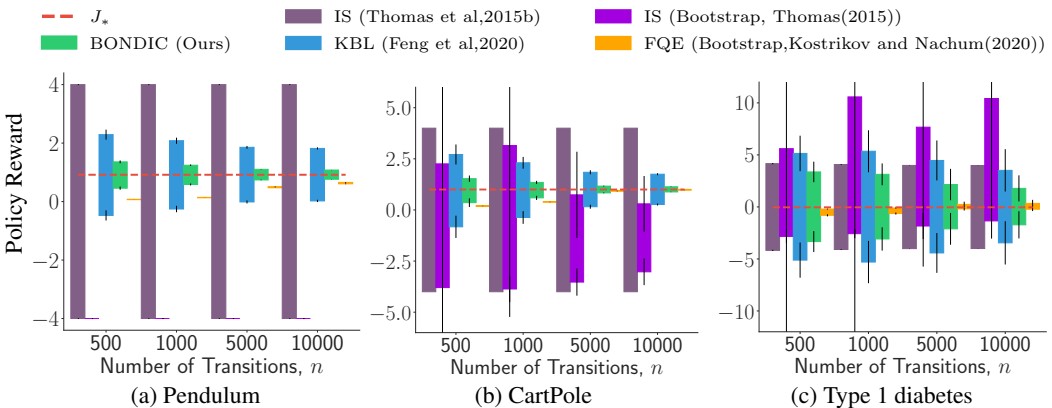

Figure 2: Results on different environments when we use a significance level of $\delta = 0.1$. The colored bars represent the confidence intervals of different methods (averaged over 50 random trials); the black error bar represents the stand derivation of the end points of the intervals over the 50 random trials.

approximately while the primal bound requires to solve a global optimization problem. Figure 1 (b) shows that we provide increasingly tight bounds as the data size $n$ increases, and the length of the interval decays with an $O(n^{-1/2})$ rate approximately. Figure 1 (c) shows that when we increase the significance level $\delta$, our bounds become tighter while still capturing the ground truth. Figure 1 (d) shows the percentage of times that the interval fails to capture the true value in a total of 100 random trials (denoted as $\hat{\delta}$) as we vary $\delta$. We can see that $\hat{\delta}$ remains close to zero even when $\delta$ is large, suggesting that our bound is very conservative. Part of the reason is that the bound is constructed by considering the worse case and we used a conservative choice of the radius $r_{\mathcal{Q}}$ and coefficient $c_{q_\pi, k}$ in Eq. (13) (See Appendix H.2).

In Figure 2 we compare different algorithms on more examples with $\delta = 0.1$. We can again see that our method provides tight and conservative interval that always captures the true value. Although FQE (Bootstrap) yields tighter intervals than our method, it fail to capture the ground truth much more often than the promised $\delta = 0.1$ (e.g., it fails in all the random trials in Figure 2 (a)).

We conduct more ablation studies on different hyper-parameter and data collecting procedure. See Appendix H.2 and H.3 for more details.

## 6   CONCLUSION

We develop a dual approach to construct high confidence bounds for off-policy evaluation with an improved rate over Feng et al. (2020). Our method can handle dependent data, and does not require a global optimization to get a valid bound. Empirical results demonstrate that our bounds is tight and valid compared with a range of existing baseline. Future directions include leveraging our bounds for policy optimization and safe exploration.

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

## A    PROOF OF THE DUAL BOUND IN THEOREM 4.2

*Proof.* Introducing a Lagrange multiplier, the bound in (8) is equivalent to

$$
\hat{J}^+_{\mathcal{Q},\mathcal{W}} = \max_{q \in \mathcal{Q}} \min_{\lambda \geq 0} \left\{ \mathbb{E}_{\mathsf{D}_{\pi,0}}[q] \; - \; \lambda \left( \max_{h \in \mathcal{W}} \frac{1}{n} \sum_{i=1}^{n} h(x_i) \hat{\mathbf{R}} q(x_i, y_i) - \varepsilon_n \right) \right\}
$$

$$
= \max_{q \in \mathcal{Q}} \min_{\lambda \geq 0} \min_{h \in \mathcal{W}} \left\{ \mathbb{E}_{\mathsf{D}_{\pi,0}}[q] \; - \; \lambda \left( \frac{1}{n} \sum_{i=1}^{n} h(x_i) \hat{\mathbf{R}} q(x_i, y_i) - \varepsilon_n \right) \right\}
$$

$$
= \max_{q \in \mathcal{Q}} \min_{\omega \in \mathcal{W}_o} \left\{ \mathbb{E}_{\mathsf{D}_{\pi,0}}[q] \; - \; \frac{1}{n} \sum_{i=1}^{n} \omega(x_i) \hat{\mathbf{R}} q(x_i, y_i) + \varepsilon_n \|\omega\|_{\mathcal{W}_o} \right\},
$$

where we use $\omega = \lambda h(x)$, such that $\lambda$ is replaced by $\|\omega\|_{\mathcal{W}_o}$. Define

$$
M(q,\,\omega;\,\hat{\mathsf{D}}_n) = \mathbb{E}_{\mathsf{D}_{\pi,0}}[q] \; - \; \frac{1}{n} \sum_{i=1}^{n} \omega(x_i) \hat{\mathbf{R}} q(x_i, y_i) + \varepsilon_n \|\omega\|_{\mathcal{W}_o}
$$

$$
= \mathbb{E}_{\hat{\mathsf{D}}_n^{\omega}}[r] + \Delta(\hat{\mathsf{D}}_n^{\omega},\,q) + \varepsilon_n \|\omega\|_{\mathcal{W}_o}.
$$

Then we have

$$
\max_{q \in \mathcal{Q}} M(q,\,\omega;\,\hat{\mathsf{D}}_n) = \mathbb{E}_{\hat{\mathsf{D}}_n^{\omega}}[r] + \max_{q \in \mathcal{Q}} \Delta(\hat{\mathsf{D}}_n^{\omega},\,q) + \varepsilon_n \|\omega\|_{\mathcal{W}_o}
$$

$$
= \mathbb{E}_{\hat{\mathsf{D}}_n^{\omega}}[r] + I_{\mathcal{Q}}(\omega;\,\hat{\mathsf{D}}_n) + \varepsilon_n \|\omega\|_{\mathcal{W}_o}
$$

$$
= \hat{F}^+_{\mathcal{Q}}(\omega).
$$

Therefore,

$$
\hat{J}^+_{\mathcal{Q},\mathcal{W}} = \max_{q \in \mathcal{Q}} \min_{\omega \in \mathcal{W}_o} M(q,\,\omega;\,\hat{\mathsf{D}}_n)
$$

$$
\leq \min_{\omega \in \mathcal{W}_o} \max_{q \in \mathcal{Q}} M(q,\,\omega;\,\hat{\mathsf{D}}_n)
$$

$$
= \min_{\omega \in \mathcal{W}_o} \hat{F}^+_{\mathcal{Q}}(\omega).
$$

The lower bound follows analogously. The strong duality holds when the Slater's condition is satisfied (Nesterov, 2013), which amounts to saying that the primal problem in (8) is convex and strictly feasible; this requires that $\mathcal{Q}$ is convex and there exists at least one solution $q \in \mathcal{Q}$ that satisfy that constraint strictly, that is, $L_{\mathcal{W}}(q;\,\hat{\mathsf{D}}_n) < \varepsilon_n$; note that the objective function $\mathcal{Q}$ is linear on $q$ and the constraint function $L_{\mathcal{W}}(q;\,\hat{\mathsf{D}}_n)$ is always convex on $q$ (since it is the sup a set of linear functions on $q$ following (3)).

$\square$

## B    PROOF OF CONCENTRATION BOUND IN THEOREM 4.1

Our proof require the following Hoeffding inequality on Hilbert spaces by Pinelis (Theorem 3, 1992); see also Section 2.4 of Rosasco et al. (2010).

**Lemma B.1.** *(Theorem 3, Pinelis, 1992) Let $\mathcal{H}$ be a Hilbert space and $\{f_i\}_{i=1}^{n}$ is a Martingale sequence in $\mathcal{H}$ that satisfies $\sup_i \|f_i\|_{\mathcal{H}} \leq \sigma$ almost surely. We have for any $\epsilon > 0$,*

$$
\Pr \left( \left\| \frac{1}{n} \sum_{i=1}^{n} f_i \right\|_{\mathcal{H}} \geq \epsilon \right) \leq 2 \exp \left( -\frac{n\epsilon^2}{2\sigma^2} \right).
$$

*Therefore, with probability at least $1 - \delta$, we have $\left\| \frac{1}{n} \sum_{i=1}^{n} f_i \right\|_{\mathcal{H}} \leq \sqrt{\frac{2\sigma^2 \log(2/\delta)}{n}}$.*

**Lemma B.2.** *Let $k(x, x')$ be a positive definite kernel whose RKHS is $\mathcal{H}_k$. Define*

$$f_i(\cdot) = \hat{\mathbf{R}}q(x_i, y_i)k(x_i, \cdot) - \mathbf{R}_\pi q(x_i)k(x_i, \cdot).$$

*Assume Assumption 2.1 holds, then $\{f_i\}_{i=1}^n$ is a Martingale difference sequence in $\mathcal{H}_k$ w.r.t. $T_{<i} := (x_j, y_j)_{j<i} \cup (x_i)$. That is, $\mathbb{E}\left[f_{i+1}(\cdot) \mid T_{<i}\right] = 0$. In addition,*

$$\left\| \frac{1}{n} \sum_{i=1}^n f_i \right\|_{\mathcal{H}_k}^2 = \frac{1}{n^2} \sum_{ij=1}^n \left( \hat{\mathbf{R}}q(x_i, y_i) - \mathbf{R}_\pi q(x_i) \right) k(x_i, x_j) \left( \hat{\mathbf{R}}q(x_j, y_j) - \mathbf{R}_\pi q(x_j) \right),$$

*and $\|f_i\|_{\mathcal{H}_k}^2 \leq c_{q,k}$ for $\forall i = 1, \ldots, n$.*

*Proof of Theorem 4.1.* Following Lemma B.1 and Lemma B.2, since $\{f_i\}_{i=1}^n$ is a Martingale difference sequence in $\mathcal{H}_k$ with $\|f_i\|_{\mathcal{H}_k} \leq c_{q,k}$ almost surely, we have with probability at least $1 - \delta$,

$$\frac{1}{n^2} \sum_{ij=1}^n \left( \hat{\mathbf{R}}q(x_i, y_i) - \mathbf{R}_\pi q(x_i) \right) k(x_i, x_j) \left( \hat{\mathbf{R}}q(x_j, y_j) - \mathbf{R}_\pi q(x_j) \right) = \left\| \frac{1}{n} \sum_{i=1}^n f_i \right\|_{\mathcal{H}_k}^2 \leq \frac{2c_{q,k} \log(2/\delta)}{n}.$$

Using Lemma B.3 below, we have

$$\left| L_\mathcal{K}(q; \hat{\mathsf{D}}_n) - L_\mathcal{K}^*(q; \hat{\mathsf{D}}_n) \right| \leq \left\| \frac{1}{n} \sum_{i=1}^n f_i \right\|_{\mathcal{H}_k} \leq \sqrt{\frac{2c_{q,k} \log(2/\delta)}{n}}.$$

This completes the proof. $\qquad\square$

**Lemma B.3.** *Assume $k(x, x')$ is a positive definite kernel. We have*

$$\left| L_\mathcal{K}(q; \hat{\mathsf{D}}_n) - L_\mathcal{K}^*(q; \hat{\mathsf{D}}_n) \right|^2 \leq \frac{1}{n^2} \sum_{ij=1}^n \left( \hat{\mathbf{R}}q(x_i, y_i) - \mathbf{R}_\pi q(x_i) \right) k(x_i, x_j) \left( \hat{\mathbf{R}}q(x_j, y_j) - \mathbf{R}_\pi q(x_j) \right).$$

*Proof.* Define

$$\hat{g}(\cdot) = \frac{1}{n} \sum_{i=1}^n \hat{\mathbf{R}}q(x_i, y_i)k(x_i, \cdot) \qquad\qquad g(\cdot) = \frac{1}{n} \sum_{i=1}^n \mathbf{R}_\pi q(x_i)k(x_i, \cdot).$$

Then we have

$$\|\hat{g}\|_{\mathcal{H}_k}^2 = \frac{1}{n^2} \sum_{ij=1}^n \hat{\mathbf{R}}q(x_i, y_i)k(x_i, x_j)\hat{\mathbf{R}}q(x_j, y_j) = \hat{L}_\mathcal{K}(q; \hat{\mathsf{D}}_n),$$

$$\|g\|_{\mathcal{H}_k}^2 = \frac{1}{n^2} \sum_{ij=1}^n \mathbf{R}_\pi q(x_i)k(x_i, x_j)\mathbf{R}_\pi q(x_j) = L_\mathcal{K}^*(q; \hat{\mathsf{D}}_n),$$

$$\|\hat{g} - g\|_{\mathcal{H}_k}^2 = \frac{1}{n^2} \sum_{ij=1}^n \left( \hat{\mathbf{R}}q(x_i, y_i) - \mathbf{R}_\pi q(x_i) \right) k(x_i, x_j) \left( \hat{\mathbf{R}}q(x_j, y_j) - \mathbf{R}_\pi q(x_j) \right).$$

The result then follows the triangle inequality $\left| \|\hat{g}\|_{\mathcal{H}_k} - \|g\|_{\mathcal{H}_k} \right| \leq \|\hat{g} - g\|_{\mathcal{H}_k}$. $\qquad\square$

## B.1 CALCULATION OF $c_{q_\pi,k}$

The practical calculation of the coefficient $c_{q_\pi,k}$ in the concentration inequality was discussed in Feng et al. (2020), which we include here for completeness.

**Lemma B.4.** *(Feng et al. (2020) Lemma 3.1) Assume the reward function and kernel function is bounded with $\sup_x |r(x)| \leq r_{\max}$ and $\sup_{x,x'} |k(x, x')| \leq K_{\max}$, we have:*

$$c_{q_\pi,k} := \sup_{x \in \mathcal{X}, y \in \mathcal{Y}} (\hat{\mathbf{R}}q_\pi(x, y))^2 k(x, x) \leq \frac{4K_{\max} r_{\max}^2}{(1 - \gamma)^2}.$$

In practice, we get access to $K_{\max}$ from the kernel function that we choose (e.g., $K_{\max} = 1$ for RBF kernels), and $r_{\max}$ from the knowledge on the environment.

## C   MORE ON THE TIGHTNESS OF THE CONFIDENCE INTERVAL

The benefit of having both upper and lower bounds is that we can empirically access the tightness of the bound by checking the length of the interval $[\hat{F}_{\mathcal{Q}}^-(\omega_-), \hat{F}_{\mathcal{Q}}^+(\omega_+)]$. However, from the theoretical perspective, it is desirable to know *a priori* that the length of the interval will decrease with a fast rate as the data size $n$ increases. We now show that this is the case if $\mathcal{W}_o$ is chosen to be sufficiently rich so that it includes a $\omega \in \mathcal{W}_o$ such that $\hat{D}_n^\omega \approx D_\pi$.

**Theorem C.1.** *Assume $\mathcal{W}_o$ is sufficiently rich to include a "good" $\omega^*$ in $\mathcal{W}_o$ with $\hat{D}_n^{\omega^*} \approx D_\pi$ in that*

$$\sup_{q \in \mathcal{Q}} \left| \mathbb{E}_{\hat{D}_n^{\omega^*}} \left[ \hat{\mathbf{R}} q(x; x', r) \right] - \mathbb{E}_{D_\pi} \left[ \hat{\mathbf{R}} q(x; x', r) \right] \right| \leq \frac{c}{n^\alpha}, \tag{19}$$

*where $c$ and $\alpha$ are two positive coefficients. Then we have*

$$\max \left\{ \hat{J}_{\mathcal{Q},\mathcal{W}}^+ - J_\pi, \ J_\pi - \hat{J}_{\mathcal{Q},\mathcal{W}}^- \right\} \leq \frac{c}{n^\alpha} + \varepsilon_n \left\| \omega^* \right\|_{\mathcal{W}_o}.$$

Assumption (19) holds if $\hat{D}_n$ is collected following a Markov chain with certain strong mixing condition and weakly converges to some limit discussion $\hat{D}_\infty$ whose support is $\mathcal{X}$, for which we can define $\omega^*(x) = D_\pi(x)/D_\infty(x)$. In this case, if $\mathcal{Q}$ is a finite ball in RKHS, then we can achieve (19) with $\alpha = 1/2$, and yields the overall bound of rate $O(n^{-1/2})$. For more general function classes, $\alpha$ depends on the martingale Rademacher complexity of function set $\hat{\mathbf{R}}\mathcal{Q} = \{\mathbf{R}q(x, y) \colon q \in \mathcal{Q}\}$ Rakhlin et al. (2015). In our empirical reults, we observe that the gap of the practically constructed bounds tend to follow the $O(n^{-1/2})$ rate.

*Proof.* Note that

$$J_\pi = \mathbb{E}_{D_\pi}[r] = \mathbb{E}_{D_\pi}[r],$$

and

$$I_{\mathcal{Q}}(\omega; \ \hat{D}_n) = \sup_{q \in \mathcal{Q}} \left\{ \mathbb{E}_{\hat{D}_n^\omega}[\gamma q(x') - q(x)] - \mathbb{E}_{D_\pi}[\gamma q(x') - q(x)] \right\}.$$

Because $\omega^* \in \mathcal{W}$, we have

$$\begin{aligned}
\hat{J}_{\mathcal{W},\mathcal{Q}}^+ - J_\pi &\leq \hat{F}_{\mathcal{Q}}^+(\omega^*) - J_\pi \\
&= \mathbb{E}_{\hat{D}_n^\omega}[r] - \mathbb{E}_{D_\pi}[r] + I_{\mathcal{Q}}(\omega_\pi; \ \hat{D}_n) + \varepsilon_n \left\| \omega^* \right\|_{\mathcal{W}_o} \\
&= \sup_{q \in \mathcal{Q}} \left\{ \mathbb{E}_{\hat{D}_n^\omega} \left[ \hat{\mathbf{R}} q(x, y) \right] - \mathbb{E}_{D_\pi} \left[ \hat{\mathbf{R}} q(x, y) \right] \right\} + \varepsilon_n \left\| \omega^* \right\|_{\mathcal{W}_o} \\
&\leq \frac{c}{n^\alpha} + \varepsilon_n \left\| \omega^* \right\|_{\mathcal{W}_o}.
\end{aligned}$$

The case of lower bound follows similarly.   □

## D   OPTIMIZATION ON $\mathcal{W}_o$

Consider the optimization of $\omega$ in $\mathcal{W}_o$

$$\hat{F}_{\mathcal{Q}}^+(\omega) := \frac{1}{n} \sum_{i=1}^n r_i \omega(x_i) + I_{\mathcal{Q}}(\omega; \ \hat{D}_n) + \left\| \omega \right\|_{\mathcal{W}_o} \sqrt{\frac{2 c_{q_\pi, k} \log(2/\delta)}{n}} \tag{20}$$

Assume $\mathcal{W}_o$ is the RKHS of kernel $k(x, \bar{x})$, that is, $\mathcal{W}_o = \mathcal{H}_k$. By the finite represener theorem of RKHS (Smola et al., 2007). the optimization of $\omega$ in RKHS $\mathcal{H}_k$ can be reduced to a finite dimensional optimization problem. Specifically, the optimal solution of (20) can be written into a form of $\omega(x) = \sum_{i=1}^n k(x, x_i) \alpha_i$ with $\|\omega\|_{\mathcal{H}_k}^2 = \sum_{i,j=1}^n k(x_i, x_j) \alpha_i \alpha_j$ for some vector $\boldsymbol{\alpha} := [\alpha_i]_{i=1}^n \in \mathbb{R}^n$. Write $\boldsymbol{K} = [k(x_i, x_j)]_{i,j=1}^n$ and $\boldsymbol{r} = [r_i]_{i=1}^n$. The optimization of $\omega$ reduces to a finite dimensional optimization on $\boldsymbol{\alpha}$:

$$\min_{\boldsymbol{\alpha} \in \mathbb{R}^n} \frac{1}{n} \boldsymbol{r}^\top \boldsymbol{K} \boldsymbol{\alpha} + I_{\mathcal{Q}}(\boldsymbol{K}\boldsymbol{\alpha}; \ \hat{D}_n) + \sqrt{\boldsymbol{\alpha} \boldsymbol{K} \boldsymbol{\alpha}} \sqrt{\frac{2 c_{q_\pi, k} \log(2/\delta)}{n}},$$

where

$$I_{\mathcal{Q}}(\boldsymbol{K\alpha}; \hat{\mathsf{D}}_n) = \max_{q \in \mathcal{Q}} \left\{ \mathbb{E}_{\mathsf{D}_{\pi,0}}[q] + \frac{1}{n}(\hat{\mathbf{T}}q)^{\top} \boldsymbol{K\alpha} \right\},$$

and $\hat{\mathbf{T}}q = [\gamma q(x'_i) - q(x_i)]_{i=1}^{n}$. When $\mathcal{Q}$ is RKHS, we can calculate $I_{\mathcal{Q}}(\boldsymbol{K\alpha}; \hat{\mathsf{D}}_n)$ using (22) in section F.

This computation can be still expensive when $n$ is large. Fortunately, our confidence bound holds for any $\omega$; better $\omega$ only gives tighter bounds, but it is not necessary to find the global optimal $\omega$. Therefore, one can use any approximation algorithm to find $\omega$, which provides a trade-off of tightness and computational cost. We discuss two methods:

**1) Approximating $\boldsymbol{\alpha}$** The length of $\boldsymbol{\alpha}$ can be too large when $n$ is large. To address this, we assume $\alpha_i = g(x_i, \theta)$, where $g$ is any parametric function (such as a neural network) with parameter $\theta$ which can be much lower dimensional than $\boldsymbol{\alpha}$. We can then optimize $\theta$ with stochastic gradient descent, by approximating all the data averaging $\frac{1}{n}\sum_{i=1}^{n}(\cdot)$ with averages over small mini-batches; this would introduce biases in gradient estimation, but it is not an issue when the goal is only to get a reasonable approximation.

**2) Replacing kernel $k$** Assume the kernel $k$ yields a random feature expansion: $k(x, \bar{x}) = \mathbb{E}_{\beta \sim \pi}[\phi(x, \beta)\phi(\bar{x}, \beta)]$, where $\phi(x, \beta)$ is a feature map with parameter $\beta$ and $\pi$ is a distribution of $\beta$. We draw $\{\beta_i\}_{i=1}^{m}$ i.i.d. from $\pi$, where $m$ is taken to be much smaller than $n$. We replace $k$ with $\hat{k}(x, \bar{x}) = \frac{1}{m}\sum_{i=1}^{m}\phi(x, \beta_i)\phi(\bar{x}, \beta_i)$ and $\mathcal{H}_k$ with $\mathcal{H}_{\hat{k}}$, That is, we consider to solve

$$\hat{J}_{\mathcal{Q},\mathcal{W}}^{+} = \min_{\omega \in \mathcal{H}_{\hat{k}}} \left\{ \hat{F}_{\mathcal{Q}}^{+}(\omega) := \frac{1}{n}\sum_{i=1}^{n} r_i \omega(x_i) + I_{\mathcal{Q}}(\omega; \hat{\mathsf{D}}_n) + \|\omega\|_{\mathcal{H}_{\hat{k}}} \sqrt{\frac{2c_{q_\pi, \hat{k}} \log(2/\delta)}{n}} \right\}.$$

It is known that any function $\omega$ in $\mathcal{H}_{\hat{k}}$ can be represented as $\omega(x) = \frac{1}{m}\sum_{i=1}^{m} w_i \phi(x, \beta_i)$, for some $\boldsymbol{w} = [w_i]_{i=1}^{m} \in \mathbb{R}^m$ and satisfies $\|\omega\|_{\mathcal{H}_{\hat{k}}}^{2} = \frac{1}{m}\sum_{i=1}^{m} w_i^2$. In this way, the problem reduces to optimizing an $m$-dimensional vector $\boldsymbol{w}$, which can be solved by standard convex optimization techniques.

# E    CONCENTRATION INEQUALITIES OF GENERAL FUNCTIONAL BELLMAN LOSSES

When $\mathcal{K}$ is a general function set, one can still obtain a general concentration bound using Radermacher complexity. Define $\hat{\mathbf{R}}q \circ \mathcal{W} := \{h(x, y) = \hat{\mathbf{R}}q(x, y)\omega(x) \colon \omega \in \mathcal{W}\}$. Using the standard derivation in Radermacher complexity theory in conjunction with Martingale theory (Rakhlin et al., 2015), we have

$$\sup_{\omega \in \mathcal{W}} \left\{ \frac{1}{n}\sum_{i=1}^{n} (\hat{\mathbf{R}}q(x_i, y_i) - \mathbf{R}_\pi q(x_i))\omega(x_i) \right\} \leq 2Rad(\hat{\mathbf{R}}q \circ \mathcal{W}) + \sqrt{\frac{2c_q \log(2/\delta)}{n}},$$

where $Rad(\hat{\mathbf{R}}q \circ \mathcal{K})$ is the sequential Radermacher complexity as defined in (Rakhlin et al., 2015). A triangle inequality yields

$$| L_k(q; \hat{\mathsf{D}}_n) - L_k(q; \hat{\mathsf{D}}_n) | \leq \sup_{\omega \in \mathcal{W}} \left\{ \frac{1}{n}\sum_{i=1}^{n} (\hat{\mathbf{R}}q(x_i, y_i) - \mathbf{R}_\pi q(x_i))\omega(x_i) \right\}$$

Therefore,

$$| L_{\mathcal{W}}(q; \hat{\mathsf{D}}_n) - L_{\mathcal{W}}(q; \hat{\mathsf{D}}_n) | \leq 2Rad(\hat{\mathbf{R}}q \circ \mathcal{W}) + \sqrt{\frac{2c_q \log(2/\delta)}{n}}, \tag{21}$$

where $c_{q,\mathcal{W}} = \sup_{\omega \in \mathcal{W}} \sup_{x,y} (\hat{\mathbf{R}}q(x, y) - \mathbf{R}_\pi q(x))^2 \omega(x)^2$. When $\mathcal{W}$ equals the unit ball $\mathcal{K}$ of the RKHS related to kernel $k$, we have $c_{q,k} = c_{q,\mathcal{W}}$, and hence this bound is strictly worse than the bound in Theorem 4.1.

# F  CLOSED FORM OF $I_{\mathcal{Q}}(\omega;\ \hat{\mathsf{D}}_n)$ WHEN $\mathcal{Q}$ IS RKHS

Similar to $L_{\mathcal{K}}(q;\ \hat{\mathsf{D}}_n)$, when $\mathcal{Q}$ is taken to be the unit ball $\tilde{\mathcal{K}}$ of the RKHS of a positive definite kernel $\tilde{k}(x,\bar{x})$, (7) can be expressed into a bilinear closed form shown in Mousavi et al. (2020):

$$I_{\mathcal{Q}}(\omega;\ \hat{\mathsf{D}}_n)^2 = A - 2B + C, \tag{22}$$

where

$$A = \mathbb{E}_{(x,\bar{x})\sim\mathsf{D}_{\pi,0}\times\mathsf{D}_{\pi,0}}\left[k(x,\bar{x})\right]$$

$$B = \mathbb{E}_{(x,\bar{x})\sim\hat{\mathsf{D}}_n^\omega\times\mathsf{D}_{\pi,0}}\left[\hat{\mathbf{T}}_\pi^x k(x,\bar{x})\right]$$

$$C = \mathbb{E}_{(x,\bar{x})\sim\hat{\mathsf{D}}_n^\omega\times\hat{\mathsf{D}}_n^\omega}\left[\hat{\mathbf{T}}_\pi^x\hat{\mathbf{T}}_\pi^{\bar{x}} k(x,\bar{x})\right],$$

were $\hat{\mathbf{T}}_\pi f(x) = \gamma f(x') - f(x)$; in $\hat{\mathbf{T}}_\pi^x\hat{\mathbf{T}}_\pi^{\bar{x}} k(x,\bar{x})$, we apply $\hat{\mathbf{T}}_\pi^{\bar{x}}$ and $\hat{\mathbf{T}}_\pi^x$ in a sequential order by treating $k$ as a function of $\bar{x}$ and then of $x$.

# G  MORE ON THE ORACLE BOUND AND ITS DUAL FORM

The oracle bound (18) provides another starting point for deriving optimization-based confidence bounds. We derive its due form here. Using Lagrangian multiplier, the optimization in (18) can be rewritten into

$$\hat{J}_{\mathcal{Q},*}^+ = \max_{q\in\mathcal{Q}}\min_\omega M(q,\omega;\ \hat{\mathsf{D}}_n), \tag{23}$$

where  $M_*(q,\omega;\ \hat{\mathsf{D}}_n) = \mathbb{E}_{\mathsf{D}_{\pi,0}}[q] - \dfrac{1}{n}\sum_{i=1}^n \omega(x_i)\left(\hat{\mathbf{R}}q(x_i,y_i) - \hat{\mathbf{R}}q_\pi(x_i,y_i)\right),$

where $\omega$ now serves as the Lagrangian multiplier. By the weak duality, we have

$$J_{\mathcal{Q},+}^* \le \hat{F}_{\mathcal{Q},*}^+(\omega) := \underbrace{\mathbb{E}_{\hat{\mathsf{D}}_n^\omega}[r] + I_{\mathcal{Q}}(\omega;\ \hat{\mathsf{D}}_n)}_{known} + \underbrace{R(\omega,\ q_\pi)}_{unknown},\quad \forall\omega.$$

and

$$R(\omega,q_\pi) = \frac{1}{n}\sum_{i=1}^n \omega(x_i)\hat{\mathbf{R}}q_\pi(x_i).$$

The derivation follows similarly for the lower bound. So for any $\omega\in\mathcal{W}_o$, we have $[\hat{J}_{\mathcal{Q},*}^-,\ \hat{J}_{\mathcal{Q},*}^+]\subseteq[\hat{F}_{\mathcal{Q},*}^-(\omega),\ \hat{F}_{\mathcal{Q},*}^+(\omega)]$.

Here the first two terms of $\hat{F}_{\mathcal{Q},*}^+(\omega)$ can be empirically estimated (it is the same as the first two terms of (16)), but the third term $R(\omega,q_\pi)$ depends on the unknown $q_\pi$ and hence need to be further upper bounded.

Our method can be viewed as constraining $\omega$ in $\mathcal{W}$, which is assumed to be the unit ball of $\mathcal{W}_o$, and applying a worst case bound:

$$\hat{F}_{\mathcal{Q},*}^+(\omega) := \mathbb{E}_{\hat{\mathsf{D}}_n^\omega}[r] + I_{\mathcal{Q}}(\omega;\ \hat{\mathsf{D}}_n) + R(\omega,\ q_\pi),\quad \forall\omega\in\mathcal{W}_o$$

$$\le \mathbb{E}_{\hat{\mathsf{D}}_n^\omega}[r] + I_{\mathcal{Q}}(\omega;\ \hat{\mathsf{D}}_n) + \|w\|_{\mathcal{W}_o}\sup_{h\in\mathcal{W}} R(h,\ q_\pi),\quad \forall\omega\in\mathcal{W}_o$$

$$\le \mathbb{E}_{\hat{\mathsf{D}}_n^\omega}[r] + I_{\mathcal{Q}}(\omega;\ \hat{\mathsf{D}}_n) + \|w\|_{\mathcal{W}_o} L_{\mathcal{W}}(q_\pi,\hat{\mathsf{D}}_n),\quad \forall\omega\in\mathcal{W}_o$$

$$\overset{w.p.1-\delta}{\le} \mathbb{E}_{\hat{\mathsf{D}}_n^\omega}[r] + I_{\mathcal{Q}}(\omega;\ \hat{\mathsf{D}}_n) + \epsilon\|w\|_{\mathcal{W}_o},\quad \forall\omega\in\mathcal{W}_o$$

$$= \hat{F}_{\mathcal{Q}}^+(\omega).$$

where the last step applies the high probability bound that $\Pr(L_{\mathcal{W}}(q_\pi,\hat{\mathsf{D}}_n)\le\varepsilon)\ge 1-\delta$. Similar derivation on the lower bound counterpart gives

$$\Pr\left(\left[\hat{F}_{\mathcal{Q},*}^-(\omega),\ \hat{F}_{\mathcal{Q},*}^+(\omega)\right]\subseteq\left[\hat{F}_{\mathcal{Q}}^-(\omega),\hat{F}_{\mathcal{Q}}^+(\omega)\right]\right)\ge 1-\delta.$$

Therefore, our confidence bound $[\hat{F}_{\mathcal{Q}}^-(\omega),\hat{F}_{\mathcal{Q}}^+(\omega)]$ is a $1-\delta$ confidence outer bound of the oracle bound $[\hat{J}_{\mathcal{Q},*}^-,\ \hat{J}_{\mathcal{Q},*}^+]\subseteq[\hat{F}_{\mathcal{Q},*}^-(\omega),\ \hat{F}_{\mathcal{Q},*}^+(\omega)]$.

**Introducing $\mathcal{Q}$ is necessarily** Our method does not require any independence assumption between the transition pairs, the trade-off is that that we have to assume that $q_\pi$ falls into a function set $\mathcal{Q}$ that imposes certain smoothness assumption. This is necessary because the data only provide information regarding $q_\pi$ on a finite number of points, and $q_\pi$ can be arbitrarily non-smooth outside of the data points, and hence no reasonable upper/lower bound can be obtained without any smoothness condition that extend the information on the data points to other points in the domain.

**Proposition G.1.** *Unless* $\Pr_{s \sim \mathsf{D}_{\pi,0}}(s \notin \{s_i, s_i'\}_{i=1}^n) = 0$, *for any* $u \in \mathbb{R}$, *there exists a function* $q \colon \mathcal{S} \times \mathcal{A} \to \mathbb{R}$, *such that*

$$\mathbb{E}_{\mathsf{D}_{\pi,0}}[q] = u, \quad \hat{\mathbf{R}}q(x_i, y_i) = \hat{\mathbf{R}}q_\pi(x_i, y_i), \quad \forall i = 1, \ldots, n.$$

*Proof.* Let $\mathcal{Q}_{\text{null}}$ be the set of functions that are zero on $\{s_i, s_i'\}_{i=1}^n$, that is,

$$\mathcal{Q}_{\text{null}} = \{g \colon \mathcal{S} \times \mathcal{A} \to \mathbb{R} \colon \ g(s, a) = 0, \ \forall s \in \{s_i, s_i'\}_{i=1}^n, \ a \in \mathcal{A}\}.$$

Then we have
$$\hat{\mathbf{R}}_\pi(q_\pi + g)(x_i, y_i) = \hat{\mathbf{R}}_\pi q_\pi(x_i, y_i), \quad \forall i = 1, \ldots, n.$$
and
$$\mathbb{E}_{\mathsf{D}_{\pi,0}}[q_\pi + g] = \mathbb{E}_{\mathsf{D}_{\pi,0}}[q_\pi] + \mathbb{E}_{\mathsf{D}_{\pi,0}}[g] = J_\pi + \mathbb{E}_{\mathsf{D}_{\pi,0}}[g].$$
Taking $g(s, a) = z\mathbb{I}(s \notin \{s_i, s_i'\}_{i=1}^n)$, where $z$ is any real number. Then we have

$$\mathbb{E}_{\mathsf{D}_{\pi,0}}[q_\pi + g] = J_\pi + z\Pr_{s \sim \mathsf{D}_{\pi,0}}(s \notin \{s_i, s_i'\}_{i=1}^n).$$

Because $\Pr_{s \sim \mathsf{D}_{\pi,0}}(s \notin \{s_i, s_i'\}_{i=1}^n). \neq 0$, we can take $z$ to be arbitrary value to make $\mathbb{E}_{\mathsf{D}_{\pi,0}}[q_\pi + g]$ to take arbitrary value. $\qquad\square$

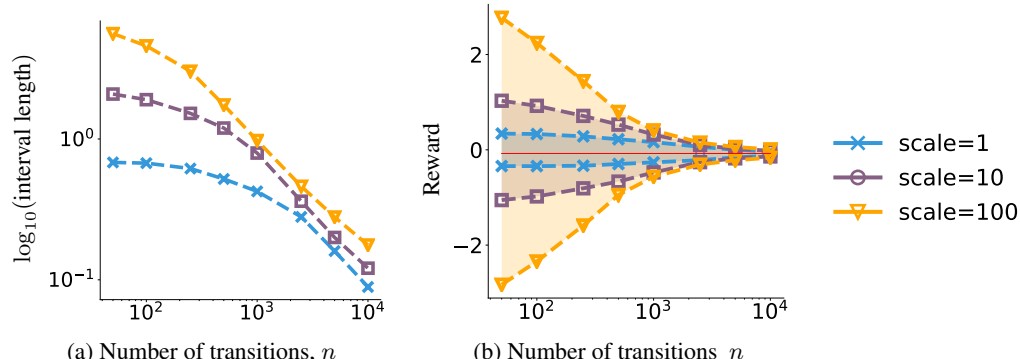

Figure 3: Ablation study on the radius $r_\mathcal{Q}$ of the function class $\mathcal{Q}$. The default collecting procedure uses a horizon length of $H = 50$. The discounted factor is $\gamma = 0.95$ by default.

## H    ABLATION STUDY AND EXPERIMENTAL DETAILS

### H.1    EXPERIMENTAL DETAILS

**Environments and Dataset Construction**    We test our method on three environments: Inverted-Pendulum and CartPole from OpenAI Gym (Brockman et al., 2016), and a Type-1 Diabetes medical treatment simulator.    For Inverted-Pendulum we discretize the action space to be $\{-1, -0.3, -0.2, 0, 0.2, 0.3, 1\}$. The action space of CartPole and the medical treatment simulator are both discrete.

**Policy Construction**    We follow a similar setup as Feng et al. (2020) to construct behavior and target policies. For all of the environments, we constraint our policy class to be a softmax policy and use PPO (Schulman et al., 2017) to train a good policy $\pi$, and we use different temperatures of the softmax policy to construct the target and behavior policies (we set the temperature $\tau = 0.1$ for target policy and $\tau = 1$ to get the behavior policy, and in this way the target policy is more deterministic than the behavior policy). We consider other choices of behavior policies in Section H.3.

For horizon lengths, We fix $\gamma = 0.95$ and set horizon length $H = 50$ for Inverted-Pendulum, $H = 100$ for CartPole, and $H = 50$ for Diabetes simulator.

**Algorithm Settings**    We test the bound in Eq.(16)-(17). Throughout the experiment, we always set $\mathcal{W} = \mathcal{K}$, a unit ball of RKHS with kernel $k(\cdot, \cdot)$. We set $\mathcal{Q} = r_\mathcal{Q}\tilde{\mathcal{K}}$, the zero-centered ball of radius $r_\mathcal{Q}$ in an RKHS with kernel $\tilde{k}(\cdot, \cdot)$. We take both $k$ and $\tilde{k}$ to be Gaussian RBF kernel. The bandwidth of $k$ and $\tilde{k}$ are selected to make sure the function Bellman loss is not large on a validation set. The radius is selected to be sufficiently large to ensure that $q_*$ is included in $\mathcal{Q}$. To ensure a sufficiently large radius, we use the data to approximate a $\hat{q}$ so that its functional Bellman loss is small than $\epsilon_n$. Then we set $r_\mathcal{Q} = 10 * \|\hat{q}\|_{\tilde{\mathcal{K}}}$. We optimize $\omega$ using the random feature approximation method described in Appendix D. Once $\omega_+$ and $\omega_-$ are found, we evaluate the bound in Eq. (16) exactly, to ensure the theoretical guarantee holds.

### H.2    SENSITIVITY TO HYPER-PARAMETERS

We investigate the sensitivity of our algorithm to the choice of hyper-parameters. The hyper-parameter mainly depends on how we choose our function class $\mathcal{Q}$ and $\mathcal{W}$.

**Radius of $\mathcal{Q}$**    Recall that we choose $\mathcal{Q}$ to be a ball in RKHS with radius $r_\mathcal{Q}$, that is,

$$\mathcal{Q} = r_\mathcal{Q}\tilde{\mathcal{K}} = \{r_\mathcal{Q}f : f \in \tilde{\mathcal{K}}\},$$

where $\tilde{\mathcal{K}}$ is the unit ball of the RKHS with kernel $\tilde{k}$. Ideally, we want to ensure that $r_\mathcal{Q} \geq \|q_*\|_{\tilde{\mathcal{K}}}$ so that $q_* \in \mathcal{Q}$.

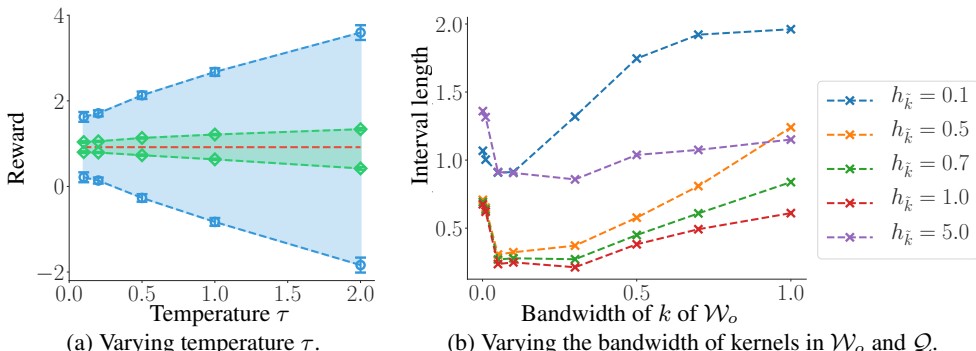

(a) Varying temperature $\tau$.  (b) Varying the bandwidth of kernels in $\mathcal{W}_o$ and $\mathcal{Q}$.

Figure 4: Ablation studies on Inverted-Pendulum. We change the temperature $\tau$ of the behavior policies in (a), and change the bandwidth of the kernel $k$ of $\mathcal{W}_o$ and the kernel $\tilde{k}$ of $\mathcal{Q}$ (denoted by $h_{\tilde{k}}$ in (b)).

Since it is hard to analyze the behavior of the algorithm when $q_*$ is unknown, we consider a synthetic environment where the true $q_*$ is known. This is done by explicitly specifying a $q_*$ inside $\tilde{\mathcal{K}}$ and then infer the corresponding deterministic reward function $r(x)$ by inverting the Bellman equation:

$$r(x) := q_*(x) - \gamma \mathbb{E}_{x' \sim \mathsf{P}_\pi(\cdot|x)}[q_*(x')].$$

Here $r$ is a deterministic function, instead of a random variable, with an abuse of notation. In this way, we can get access to the true RKHS norm of $q_*$:

$$\rho^* = \|q_*\|_{\tilde{\mathcal{K}}}.$$

For simplicity, we set both the state space $\mathcal{S}$ and action space $\mathcal{A}$ to be $\mathbb{R}$ and set a Gaussian policy $\pi(a|s) \propto \exp(f(s,a)/\tau)$, where $\tau$ is a positive temperature parameter. We set $\tau = 0.1$ as target policy and $\tau = 1$ as behavior policy.

Figure 3 shows the results as we set $r_{\mathcal{Q}}$ to be $\rho^*$, $10\rho^*$ and $100\rho^*$, respectively. We can see that the tightness of the bound is affected significantly by the radius when the number $n$ of samples is very small. However, as the number $n$ of samples grow (e.g., $n \geq 2 \times 10^3$ in our experiment), the length of the bounds become less sensitive to the changing of the predefined norm of $\mathcal{Q}$.

**Similarity Between Behavior Policy and Target Policy**  We study the performance of changing temperature of the behavior policy. We test on Inverted-Pendulum environment as previous described. Not surprisingly, we can see that the closer the behavior policy to the target policy (with temperature $\tau = 0.1$), the tighter our confidence interval will be, which is observed in Figure 4(a).

**Bandwidth of RBF kernels**  We study the results as we change the bandwidth in kernel $k$ and $\tilde{k}$ for $\mathcal{W}$ and $\mathcal{Q}$, respectively. Figure 4(b) shows the length of the confidence interval when we use different bandwidth pairs in the Inverted-Pendulum environment. We can see that we get relatively tight confidence bounds as long as we set the bandwidth in a reasonable region (e.g., we set the bandwidth of $k$ in $[0.1, 0.5]$, the bandwidth of $\tilde{k}$ in $[0.5, 3]$).

## H.3 SENSITIVITY TO THE DATA COLLECTION PROCEDURE

We investigate the sensitivity of our method as we use different behavior policies to collect the dataset $\hat{\mathsf{D}}_n$.

**Varying Behavior Policies**  We study the effect of using different behavior policies. We consider the following cases:

1. Data is collected from a single behavior policy of form $\pi_\alpha = \alpha\pi + (1-\alpha)\pi_0$, where $\pi$ is the target policy and $\pi_0$ is another policy. We construct $\pi$ and $\pi_0$ to be Gaussian policies of form $\pi(a|s) \propto \exp(f(s,a)/\tau)$ with different temperature $\tau$, where temperature for target policy is $\tau = 0.1$ and temperature for $\pi_0$ is $\tau = 1$.

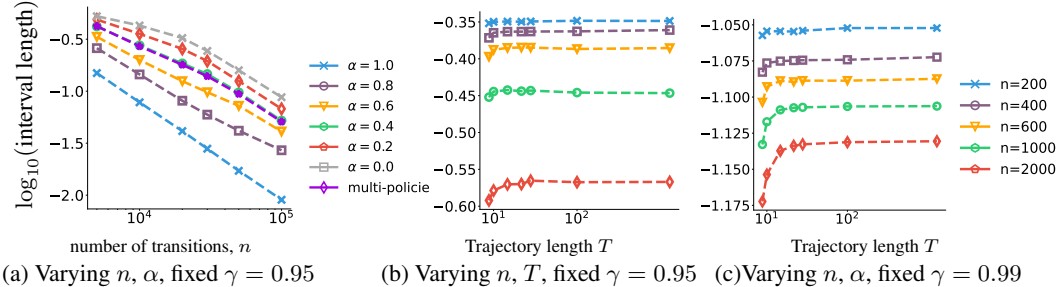

(a) Varying $n$, $\alpha$, fixed $\gamma = 0.95$      (b) Varying $n$, $T$, fixed $\gamma = 0.95$    (c)Varying $n$, $\alpha$, fixed $\gamma = 0.99$

Figure 5: Ablation studies on the data collection procedure, as we (a) change the behavior policies, and (b)-(c) change the trajectory lengths. The other settings are the same as that in Figure 3.

2. The dataset $\hat{D}_n$ is the combination of the data collected from multiple behavior policies of form $\pi_\alpha$ defined as above, with $\alpha \in \{0.0, 0.2, 0.4, 0.6, 0.8\}$.

We show in Figure 5(a) that the length of the confidence intervals by our method as we vary the number $n$ of transition pairs and the mixture rate $\alpha$. We can see that the length of the interval decays with the sample size $n$ for all mixture rate $\alpha$. Larger $\alpha$ yields better performance because the behavior policies are closer to the target policy.

**Varying Trajectory Length $T$ in $\hat{D}_n$**    As we collect $\hat{D}_n$, we can either have a small number of long trajectories, or a larger number of short trajectories. In Figure 5(b)-(c), we vary the length $T$ of the trajectories as we collect $\hat{D}_n$, while fixing the total number $n$ of transition pairs. In this way, the number of trajectories in each $\hat{D}_n$ would be $m = n/T$. We can see that the trajectory length does not impact the results significantly, especially when the length is reasonably large (e.g., $T \geq 20$).

# I   More Related Works

We give a more detailed overview of different approaches for uncertainty estimation in OPE.

**Finite-Horizon Importance Sampling (IS)**   Assume the data is collected by rolling out a known behavior policy $\pi_0$ up to a trajectory length $T$, then we can estimate the finite horizon reward by changing $\mathbb{E}_{\pi,\mathsf{P}}[\cdot]$ to $\mathbb{E}_{\pi_0,\mathsf{P}}[\cdot]$ with importance sampling(e.g., Precup et al., 2000; Precup, 2001; Thomas et al., 2015a;b). Taking the trajectory-wise importance sampling as an example, assume we collect a set of independent trajectories $\tau_i := \{s_t^i, a_t^i, r_t^i\}_{t=0}^{T-1}$, $i = 1, \ldots, m$ up to a trajectory length $T$ by unrolling a known *behavior policy* $\pi_0$. When $T$ is large, we can estimate $J_*$ by a weighted averaging:

$$\hat{J}^{\text{IS}} = \frac{1}{m} \sum_{i=1}^{m} \omega(\tau_i) J(\tau_i), \quad \text{where} \quad \omega(\tau_i) = \prod_{t=0}^{T-1} \frac{\pi(a_t^i | s_t^i)}{\pi_0(a_t^i | s_t^i)}, \quad J(\tau_i) = \sum_{t=0}^{T-1} \gamma^t r_t^i. \quad (24)$$

One can construct non-asymptotic confidence bounds based on $\hat{J}^{\text{IS}}$ using variants of concentration inequalities (Thomas, 2015; Thomas et al., 2015b). Unfortunately, a key problem with this IS estimator is that the importance weight $\omega(\tau_i)$ is a product of the density ratios over time, and hence tends to cause an explosion in variance when the trajectory length $T$ is large. Although improvement can be made by using per-step and self-normalized weights (Precup, 2001), or control variates (Jiang & Li, 2016; Thomas & Brunskill, 2016), the *curse of horizon* remains to be a key issue to the classical IS-based estimators (Liu et al., 2018a).

Moreover, due to the time dependency between the transition pairs inside each trajectory, the non-asymptotic concentration bounds can only be applied on the trajectory level and hence decay with the number $m$ of independent trajectories in an $O(1/\sqrt{m})$ rate, though $m$ can be small in practice. We could in principle apply the concentration inequalities of Markov chains (e.g., Paulin, 2015) to the time-dependent transition pairs, but such inequalities require to have an upper bound of certain mixing coefficient of the Markov chain, which is unknown and hard to construct empirically. Our work addresses these limitations by constructing a non-asymptotic bound that decay with the number $n = mT$ of transitions pairs, while without requiring known behavior policies and independent trajectories.

**Infinite-Horizon, Behavior-Agnostic OPE**   Our work is closely related to the recent advances in infinite-horizon and behavior-agnostic OPE, including, for example, Liu et al. (2018a); Feng et al. (2019); Tang et al. (2020a); Mousavi et al. (2020); Liu et al. (2020); Yang et al. (2020b); Xie et al. (2019); Yin & Wang (2020), as well as the DICE-family (e.g., Nachum et al., 2019a;b; Zhang et al., 2020a; Wen et al., 2020; Zhang et al., 2020b). These methods are based on either estimating the value function, or the stationary visitation distribution, which is shown to form a primal-dual relation (Tang et al., 2020a; Uehara et al., 2020; Jiang & Huang, 2020) that we elaborate in depth in Section 3.

Besides Feng et al. (2020) which directly motivated this work, there has been a recent surge of interest in interval estimation under infinite-horizon OPE (e.g., Liu et al., 2018b; Jiang & Huang, 2020; Duan et al., 2020; Dai et al., 2020; Feng et al., 2020; Tang et al., 2020b; Yin et al., 2020; Lazic et al., 2020). For example, Dai et al. (2020) develop an asymptotic confidence bound (CoinDice) for DICE estimators with an i.i.d assumption on the off-policy data; Duan et al. (2020) provide a data dependent confidence bounds based on Fitted Q iteration (FQI) using linear function approximation when the off-policy data consists of a set of independent trajectories; Jiang & Huang (2020) provide a minimax method closely related to our method but do not provide analysis for data error; Tang et al. (2020b) propose a fixed point algorithm for constructing deterministic intervals of the true value function when the reward and transition models are deterministic and the true value function has a bounded Lipschitz norm.

**Model-Based Methods**   Since the model $\mathsf{P}$ is the only unknown variable, we can construct an estimator $\hat{\mathsf{P}}$ of $\mathsf{P}$ using maximum likelihood estimation or other methods, and plug it into Eq. (1) to obtain a plug-in estimator $\hat{J} = J_{\pi,\hat{\mathsf{P}}}$. This yields the model-based approach to OPE (e.g., Jiang & Li, 2016; Liu et al., 2018b). One can also estimate the uncertainty in $J_{\pi,\hat{\mathsf{P}}}$ by propagating the uncertatinty in $\hat{\mathsf{P}}$ (e.g., Asadi et al., 2018; Duan et al., 2020), but it is hard to obtain non-asymptotic

and computationally efficient bounds unless $\hat{\mathsf{P}}$ is assumed to be simple linear models. In general, estimating the whole model $\mathsf{P}$ can be an unnecessarily complicated problem as an intermediate step of the possibly simpler problem of estimating $J_{\pi,\mathsf{P}}$.

**Bootstrapping, Bayes, Distributional RL**   As a general approach of uncertainty estimation, bootstrapping has been used in interval estimation in RL in various ways (e.g., White & White, 2010; Hanna et al., 2017; Kostrikov & Nachum, 2020; Hao et al., 2021). Bootstrapping is simple and highly flexible, and can be applied to time-dependent data (as appeared in RL) using variants of block bootstrapping methods (e.g., Lahiri, 2013; White & White, 2010). However, bootstrapping typically only provides asymptotic guarantees; although non-asymptotic bounds of bootstrap exist (e.g., Arlot et al., 2010), they are sophistic and difficult to use in practice and would require to know the mixing condition for the dependent data. Moreover, bootstrapping is time consuming since it requires to repeat the whole off-policy evaluation pipeline on a large number of resampled data.

Bayesian methods (e.g., Engel et al., 2005; Ghavamzadeh et al., 2016b; Yang et al., 2020a) offer another general approach to uncertainty estimation in RL, but require to use approximate inference algorithms and do not come with non-asymptotic frequentist guarantees. In addition, distributional RL (e.g., Bellemare et al., 2017) seeks to quantify the intrinsic uncertainties inside the Markov decision process, which is orthogonal to the epistemic uncertainty that we consider in off-policy evaluation.

