# OpenReview forum: "Non-asymptotic Confidence Intervals of Off-policy Evaluation:  Primal and Dual Bounds "
_ICLR.cc/2021/Conference — ICLR 2021 Poster_

### Official Review · AnonReviewer4 · 2020-10-27
**This paper derives a tighter non-asymptotic confidence interval for off-policy evaluation.**

**Rating:** 7
**Confidence:** 3

**Review:**

This paper proposes an approach to construct confidence intervals using finite samples for off-policy evaluation. The paper improves the bound of a previous paper from $O(n^{-1/4})$ to $O(n^{-1/2})$ and avoids solving global optimum by introducing the dual. It is also noted that the results do not only apply to independent data. The authors further show the advantage of their method as compared to existing baselines in simulations, where their approach demonstrates good coverage and tight bound. The paper is well written.

I have some comments/thoughts as below:
- how would point estimation of the policy value be derived using such an approach? In many cases, it's also desirable to give a point estimation so that we can compute mse , etc.
- it may be worth mentioning the findings (or some intuition) of the ablation study in Appendix H in the main body to be more educational, such as how the overlap between behavior policy and target policy influences the results.

Please address and clarify these points above.

---

> ### Author Response · Authors · 2020-11-21
> **Reply to R#4**
>
> Thank you for your positive comments and valuable suggestions. Here are the responses for your questions.
>
> - [Q1: Can we derive a point estimation method using this approach?]
>
> The point estimation counterpart of our method has in fact been developed in the infinite horizon OPE literature and is reviewed in Section 3 of our paper as background. See also the related work paragraph in Section 1 and Appendix I as well.
>
> - [Q2: It is worth mentioning the findings in ablation to the main text.]
>
> That’s a good suggestion. We have updated the paper and mentioned the ablation results in Section 5 in our revision.

---

### Official Review · AnonReviewer3 · 2020-10-27
**An improved confidence interval construction for a special case of off-policy evaluation in MDPs**

**Rating:** 6
**Confidence:** 4

**Review:**

**General overview**
The paper studies an off-policy evaluation (OPE) problem for Markov decision processes (MDPs). It suggests an optimization-based method that can construct a non-asymptotic confidence interval, for a given confidence level, for the value function of a policy starting from a fixed initial distribution. The paper builds on the works of Feng et al. (2019, 2020); the main advantages of the current work with respect to the previous methods are that the suggested approach guarantees a faster convergence rate, it does not require full independence between transition pairs, and it does not need the global optimal solution of the underlying optimization problem, in order to construct guaranteed confidence intervals. The authors present some theoretical results about the construction, including a discussion on the special case of using RKHS approaches, and also present numerical experiments on benchmark problems, such as the inverted-pendulum, cartpole and type-1 diabetes.

**Strengths of the paper**
-- In general, constructing confidence regions for value functions of RL policies is an important problem (however, the paper only addresses a restrictive special case of this problem, see below).
-- The presented method is a clear improvement over a recent OPE confidence interval construction with fewer conditions and better rate (for this special case of OPE).
-- The properties of the method are analyzed theoretically, "primal" and "dual" bounds are given.
--  Illustrative numerical experiments are also presented on benchmark RL problems.

**Weaknesses of the paper**
-- The paper is obscurely written, for example, several objects are not precisely defined. It is not clear from the description on page 2 whether the state and action spaces of the MDP are finite or they can be more general (for example, Borel spaces). If the state space is finite, then using RKHS approaches (at least theoretically) seems unnecessary. On the other hand, if the state space can be infinite, then some structural assumptions are needed, for example, about its measurability.
-- It is also not clear how should the quantity I_Q(omega, hat{D}_n) computed in practice.
-- The precise interpretation of the theoretical results, such as Theorem 4.2, is not obvious, either.
-- A major drawback of this work is that it only considers the OPE problem from a fixed, known initial distribution of the states. This is no more general than solving the problem for only one particular starting state. A much more interesting problem would be to have a confidence region for the entire value function, under some structural assumptions on the problem.
-- The claim in the "Experiments" part that "Our method is safe (always captures the true value) and tight [...], while the other methods are either too lose or often fail to capture the ground truth" is dubious, as the goal (see also on page 2) is not to "always" capture the true value, but to capture it *with a given probability*. Also, increasing this probability will make the resulting interval less tight. Mathematically, the Type I and II errors are traded off against each other.

**Minor comments**
-- Some more explanations and motivations would be needed for the concept of "functional" Bellman loss.
-- In the sentence below equation (8) "sup" and "inf" should be used instead of "min" and "max" (or some argument should be given that the maximum and minimum can be actually obtained).
-- It would be better to cite the 2018 extended 2nd edition of Sutton and Barto's classical RL book, instead of its 1st edition published in 1998.

---

> ### Author Response · Authors · 2020-11-21
> **Reply to R#3**
>
> Thank you for your valuable comments and suggestions. Here are responses for your questions.
>
> - [Q1: Not clear if state/action is finite and structure assumption]
>
> The state space can be any measurable space on which RKHS can be defined, it can be, for example, a countable set or typical Euclidean space R^d. We will clarify this in the paper. Since the main goal of the paper is proposing a new algorithm, we do not want to bring up abstract measure theoretic issues.
> It is useful to use RKHS for discrete space, because when the space is high dimensional and combinatorially large, RKHS provides a small set of functions that can be handled computationally efficiently, by using kernels based on some notion of distance (such as hamming distance).
>
> - [Q2: How to compute $I_Q$ in practice?]
>
> The computation of $I_Q$ is shown in Appendix D due to space limit. Reviewer can also refer to Mousavi’2020 Equation (11) (though they focus on the average case with $\gamma=1$).
>
> - [Q3: A major drawback of this work is that it only considers the OPE problem from a fixed, known initial distribution of the states. Can this method be applied to the confidence region for the entire value function?]
>
> The OPE problem with a fixed initial distribution is a standard setting studied in many works (Jiang et. al 2016, Thomas et.al, 2016, Liu et.al, 2018). It is already a very challenging problem to provide non-asymptotic interval estimation under this setting, for which we provide a new efficient tool. We believe that our work provides a basis for various further extensions, including the problem of estimating the value functions as the reviewer suggested.
>
>
> - [Q4: The wording of “safe” in experiment and other minor wording issues.]
>
> Thanks for pointing out. We have rephrased the wording in our revision.
>
> [reference]
> 1. Jiang, Nan, and Li, Lihong. "Doubly robust off-policy value evaluation for reinforcement learning." ICML, 2016.
> 2. P. S. Thomas and E. Brunskill. Data-Efficient Off-Policy Policy Evaluation for Reinforcement Learning. ICML, 2016.
> 3. Liu, Qiang, et al. "Breaking the curse of horizon: Infinite-horizon off-policy estimation." Neurips. 2018.

---

### Official Review · AnonReviewer2 · 2020-10-28
**A potentially practical method for OPE**

**Rating:** 7
**Confidence:** 4

**Review:**

The objective of this paper is to provide a method to produce tighter confidence intervals for off-policy evaluation. The paper claims to develop a new primal-dual perspective on OPE confidence intervals and a tight concentration inequality. It develops both theoretical and empirical evidence to support its claims.

Previous methods (Feng et al. 2020) estimate the high upper confidence bound on the bellman residual for q_pi given a set of data and then perform a global optimization procedure to find the largest q function with an empirical error less than the upper bound on the residual. This paper proposes to estimate instead of a confidence interval for the expected bellman error over the empirical data set for any q function. The dual approach from other OPE estimators is then leveraged to create high confidence bounds on the objective function.

This paper's strengths are that the presented method could significantly improve confidence intervals for off-policy evaluation with a moderate length horizon and when an RKHS can represent the q function. There is both theory and empirical data to gain insights into the effectiveness of the method and show it a possible solution.

Although the method appears to be effective, I cannot yet recommend it for acceptance due to some of the unsubstantiated claims and a lack of clarity in the paper's writing. There are also some ways that the experiments should be improved.

This paper claims to produce a tight concentration inequality, but this is not proven. The claim may be a confusion of the wording and that it is intended to mean that the presented method is only a relative improvement over existing methods. Can the authors clarify the intended scope of this claim? If the claim is to be a tight concentration inequality, then a proof showing it cannot be improved is required.

Additionally, it is stated that this work is a "substantial extension of [dual form OPE] to the non-asymptotic region, and therefore is both of theoretical and practical significance." However, it is unclear what problem this paper overcomes in previous methods to make this a substantial extension to the non-asymptotic region. The formulation and the use of the dual form do not appear substantial as it is currently presented because, as the authors point out, many others have proposed this form. What is the source of this substantial extension?

In the definition of c_{q,k}, the supremum over x,y is used, but it is unclear if this is over the empirical data or any possible x,y. Can the authors clarify this?

Notes about experiments:
The results look very promising for the method, and the ablation studies in the appendix help understand some of its properties. However, there is significant room for improvement in experimental design. The main component lacking in the experiments is a demonstration of the limitations of the method. The only thing I can take away from these results is that this method worked on these problems. I do not doubt that this method is more effective than PDIS for moderate length horizons, but cannot predict when it will be useful.

Horizons of length 50 and 100 were used, but the discount factor was set to 0.95, making the effective horizon only 20. I do not see why this is an effective choice for demonstrating the capabilities of the method. Furthermore, all of these environments are typically simulated with much longer horizons (at least a thousand steps for cart-pole, inverted pendulum, and the diabetes simulator). It would be helpful to see this method's capabilities in a more typical experimental setup.

Another shortcoming of the experiments is that the behavior policy is only a high-temperature version of the evaluation policy. Typically, when off-policy estimation is performed, it is not to reduce the policy's noise but evaluate a different policy altogether. Since this work makes no claims or assumptions about the policy used to generate the data, it would make sense to demonstrate that the confidence intervals are accurate and reliable when using significantly different behavior policies or multiple behavior policies.


Writing notes:
Overall, the paper's writing indicates that it was written for experts who already know and understand the paper's concepts. It would be more useful to the ICLR and RL communities if the paper were written for a more general audience.

Minor notes:
In Section 2, the objective function is called the expected reward, which implies an average reward setting, but this is not the objective function's formulation. The wording is confusing here.

There is a missing reference to proof of theorem 4.2 in the appendix.
The term IS is used for importance sampling, but the formulation is actually per decision importance sampling (PDIS). Specifying this would add clarity to the paper.

Figure 1 (c) is not described.

--EDIT-- updated score to 7 after the author's response to questions and changes to the paper.

---

> ### Author Response · Authors · 2020-11-21
> **Reply to R#2**
>
> We thank the reviewer for your valuable comments. The followings are our response to your questions:
>
> - [Q1: Clarification on the main strength of our paper]
>
> We want to clarify that the main theoretical contribution of our paper is: 1) improved the rate of concentration inequality to Feng’s paper; 2) Our non-asymptotic bound does not require i.i.d. assumption over transition pairs; 3) A practical algorithm leverages the dual property that always provides valid bound but does not require global optimization.
>
> - [Q2: Relative improvement rather than the *tightest* bound]
>
> We mean our bound is *tighter* than the inequality in Feng et al. 2020. We have clarified this in the paper.
>
> - [Q3: (Last sentence in related work part.) What is the source of this substantial extension?]
>
>  The substantial extension is to make it work for finite sample cases. We have rephrased this sentence in the revision.
>
> - [Q4: definition of coefficient $c_{q,k}$?]
>
> The supremum is defined on the domain of all x,y values, which can be calculated practically by the upper bound of the reward function and kernel function. We added detailed information for calculating $c_{q,k}$ in Appendix B.1. See also Lemma 3.1 of (Feng et al. 20).
>
> - [Q5: A demonstration of the limitation of this method]
>
> Our method provides conservative, non-asymptotic bounds, but can have longer interval length than asymptotic methods such as FQE+Bootstrap. A direction worth efforts is to further improve the tightness without satisfying the non-asymptotic property.
>
> - [Q6: Horizon length and discounted factor design in the experiment]
>
> We want to clarify that in the infinite horizon OPE, each transition data $(s,a,s’,r)$ is equally important since we are not using a trajectory-based estimator. A transition pair even at time step 200 can be contributed a lot once the state is “close” to the stationary distribution of $d_\pi$. See Liu et al. 2018 Appendix B for more illustration.
>
> We add an ablation study on changing the horizon length when we collect the data in our revision. See Figure 5(b) in Appendix H. We can see that the result is not  heavily influenced by the horizon length. On the other hand, as shown in Figure 1(b), the length of our confidence interval tends to decay with the number of total transition pairs with a $O(n^{-1/2})$ rate.
>
> - [Q7: Choice of behavior policy]
>
> We add an ablation study on the choice of behavior policy in our revision (see Figure 5(b) in Appendix H). We can see that even under multiple behavior policies scenario, our method can still yield safe and tight bound compared to single behavior policy.
>
> - [Q8: Written for experts?]
>
> The technical nature of this work and the 8 page long limitation makes the writing challenging. We will further improve the clarity of the paper. Please let us know if the reviewer has specific suggestions.
>
> - [Q9: Minor wording issues]
>
> We thank the reviewer for pointing out the wording issue and we updated them accordingly in our revision.
>
> [reference]
> 1. Feng, Yihao, et al. “Accountable off-policy evaluation with kernel bellman statistics.” ICML 2020.
> 2. Liu, Qiang, et al. "Breaking the curse of horizon: Infinite-horizon off-policy estimation." Neurips. 2018.

---

> > ### Comment · AnonReviewer2 · 2020-11-22
> > **Response to author's response**
> >
> > Thank you for your detailed response and the changes to the paper. My largest complaints have been addressed, and I will now recommend the paper for acceptance.
> >
> > Additionally, Section H's ablation studies are much more informative. I have an additional question on the horizon length experiments, though. As stated, the confidence interval's length does not change significantly after a horizon of at least 20. However, this suggests that the confidence interval is limited by the discount factor, set to 0.95. My original complaint with the experiments was that the discount factor for the environments was too low. For example, if a user of the method wants to know if it is likely that a new policy can balance Cart-Pole for 1000 time steps, a discount factor of 0.95 is too low. If the experiments used a discount factor to better capture performance for long-horizon tasks, the experiments would clearly show how well this method will perform in practice.

---

### Official Review · AnonReviewer1 · 2020-10-29
**Review for Non-asymptotic Confidence Intervals of Off-policy Evaluation: Primal and Dual Bounds**

**Rating:** 8
**Confidence:** 3

**Review:**

This work constructs non-asymptotic confidence intervals for off-policy evaluation. This is achieved by assuming that the reward at any given time only depends on the state action pair, leveraging that assumed structure to define the difference between the empirical and estimated bellman residual operators as a Martingale difference sequence. This, in turn, then allows the authors to apply a Hoeffding-like concentration inequality which applies to Hilbert spaces. The authors then provide a derivation of the confidence bounds by considering the divergence between policies. The work improves on the rate of prior work from $O(n^{-\frac{1}{4}})$ to $O(n^{-\frac{1}{2}})$ and allows for estimation without the need of global optimality via the dual formulation, both of which are very nice additions to the literature. Experimental evaluation backs up the authors’ claims, showing very strong performance with respect to prior art.


I found this paper to be very well written and presented, with impressively thorough theoretical results and good empirical validation.
A couple of minor questions:

(1) Performance of the proposed method when the functions don’t lie in an RKHS. It appears that the formulation in appendix E provides a bound which uses Rademacher complexity and doesn’t rely on an RKHS. Can the authors provide intuition around how much worse we would expect this to be in practice?

(2) Proposition G.1 makes a case for the necessity of assuming a smoothness condition in the absence of an independence between transition pairs. Under a milder condition on transition pair independence, e.g. a mixing condition, are similar bounds to those presented in the current work attainable?

---

> ### Author Response · Authors · 2020-11-21
> **Reply to R#1**
>
> Thank you for your positive and detailed comments. The followings are the responses:
>
> - [Q1: Rademacher complexity of non RKHS case, how much worse?]
>
> If the function space is RKHS, our bound is tighter than the one given by Rademacher complexity, so our bound is preferred. When the function space is not RKHS, e.g. neural networks, the Rademacher complexity should be used and we will need to use a proper upper bound of Rademacher complexity, whose tightness decides the quality of the final bound.
>
> - [Q2: Proposition G.1 assumption?]
>
> If we assume the transition pairs are (weakly) independent, typical importance sampling can be used without requiring the smoothness condition, but it would require to know the behavior policy and suffers from the curse of horizon.

---

### Author Response · Authors · 2020-11-21
**General Response**

We thank all reviewers for their valuable comments and suggestions. We submitted a new version and highlight the changing part in red with the following revisions:
1. We have added Section H.3 of ablation study in Appendix H on changing the source of historical data by (a) changing the data collecting horizon length and (b) changing the behavior policies (suggested by reviewer #2). And we have briefly mentioned the ablation study results in main text at the end of Section 5 (suggested by reviewer #4).
2. We have added a Section B.1 to explain how to choose the coefficient $c_{q_\pi,k}$ in practice.
3. We have changed the wording “tight bound” to “tighter bound” in Section 4.1 to avoid misleading (suggested by reviewer #2).
4. We have added an explanation at the end of the related work to explain why our work is an “substantial extension” of previous methods (suggested by reviewer #2).
5. We have rephrased the explanation of wording “safe” in experimental part (suggested by reviewer #3)
6. We have fixed the other minor typos, wording and citation issues.

---

### Decision · Program_Chairs · 2021-01-07
**Final Decision**

**Decision:**

Accept (Poster)

**Comment:**

The paper introduces new tighter non-asymptotic confidence intervals for off-policy evaluation, and all reviewers generally liked the results. I recommend acceptance of this paper. Some concerns of Reviewer2 and Reviewer3 are not fully addressed in your rebuttal. Please make sure to address all remaining issues.